# Constrained Discrete Diffusion

**Michael Cardei**[*]
University of Virginia
ntr2rm@virginia.edu

**Jacob K. Christopher**[*]
University of Virginia
csk4sr@virginia.edu

**Thomas Hartvigsen**
University of Virginia
hartvigsen@virginia.edu

**Bhavya Kailkhura**
Lawrence Livermore National Laboratory
kailkhura1@llnl.gov

**Ferdinando Fioretto**[†]
University of Virginia
fioretto@virginia.edu

## Abstract

Discrete diffusion models are a class of generative models that construct sequences by progressively denoising samples from a categorical noise distribution. Beyond their rapidly growing ability to generate coherent natural language, these models present a new and important opportunity to enforce sequence-level constraints, a capability that current autoregressive models cannot natively provide. This paper capitalizes on this opportunity by introducing *Constrained Discrete Diffusion* (CDD), a novel integration of differentiable constraint optimization within the diffusion process to ensure adherence to constraints, logic rules, or safety requirements for generated sequences. Unlike conventional text generators that often rely on post-hoc filtering or model retraining for controllable generation, CDD directly imposes constraints into the discrete diffusion sampling process, resulting in a training-free and effective approach. Experiments in toxicity-controlled text generation, property-constrained molecule design, and instruction-constrained text completion demonstrate that CDD achieves *zero constraint violations* in a diverse array of tasks while preserving fluency, novelty, and coherence, while outperforming autoregressive and existing discrete diffusion approaches.

## 1 Introduction

Language generation, alongside many scientific problems, admits a natural representation as the generation of a discrete sequence from a finite alphabet. Examples range from natural language to molecular SMILES strings and linearized chemical procedures [1–4]. While large language models (LLMs) and related sequence transformers have recently accelerated discovery by proposing candidate sequences with desirable attributes, their autoregressive sampling mechanism produces tokens sequentially, hindering the ability to provide a native mechanism to ensure constrained feasibility. When these constraints are not satisfied, the generated outputs may be unreliable and ineffective in real-world applications. For example, a single out-of-vocabulary atom in a SMILES string can render a synthesized compound meaningless or even dangerous; similarly an overlooked volume unit in an autonomous laboratory protocol can trigger unsafe reactions.

To limit such risks, LLMs are often deployed with a variety of guardrails. These include soft alignment through reinforcement learning from human feedback (RLHF), rejection sampling, or heuristic post-processing [3, 5]. However, these methods do not offer provable compliance with logical formulas, stoichiometric rules, or regulatory thresholds. This is exacerbated by the autoregressive nature of

---

[*]Equal contribution.
[†]Contact author.

39th Conference on Neural Information Processing Systems (NeurIPS 2025).

common language models, which generate sequences one token at a time, making it difficult to enforce constraints at the sequence level.

In contrast, discrete diffusion models offer a compelling alternative generative mechanism [6–9]. They refine a fully corrupted sequence by iteratively denoising the entire sample, and have shown strong performance across a variety of tasks, including text generation, molecular design, and program synthesis [4, 7, 10]. Since each step exposes a global view of the partial sequence, it creates a natural opportunity to impose sequence-level structure. This work introduces **Constrained Discrete Diffusion** (CDD), a framework that capitalizes on this property by coupling discrete diffusion with a differentiable projection operator. Given a corrupted sequence at a particular diffusion step, the projection searches for a new candidate that stays close to the model's current score distribution,

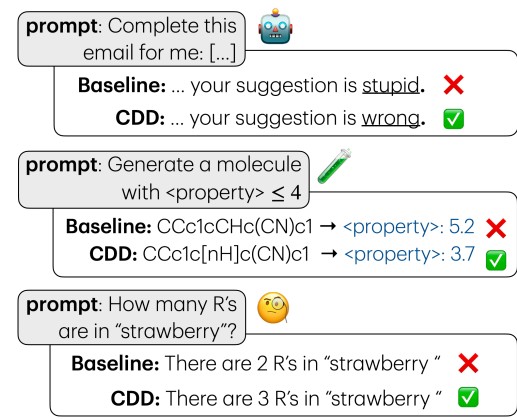

Figure 1: Comparison of **Constrained Discrete Diffusion** and baseline models. CDD imposes constraints without sacrificing fluency or expressiveness.

preserves entropy-based diversity, and simultaneously satisfies all user-defined constraints before the next denoising update. Because the introduced projection acts only at sampling time, the method is *training-free* and needs neither model retraining nor post hoc filtering. Its stochastic formulation also preserves the generative diversity, which is crucial for exploratory scientific design, as will be shown in the empirical analysis.

Specifically, CDD is evaluated on three representative use cases illustrated in Figure 1: (i) toxicity mitigation with tunable thresholds for safe LLM deployment, (ii) property adherence for molecular sequence generation, and (iii) instruction-following tasks requiring precise symbolic constraints. The experimental results demonstrate *zero threshold violations* for toxicity, up to a *203.4% increase* in novel molecule generation with complete property adherence, and *100% satisfaction* for instruction-following constraints.

**Contributions.** This paper provides the following contributions:

1. It introduces Constrained Discrete Diffusion (CDD), a novel framework that integrates discrete diffusion models with a differentiable projection operator, enforcing global sequence-level constraints directly within the diffusion sampling process.
2. It formulates this projection operator by solving a Lagrangian dual program at each denoising step, making the constrained optimization tractable and effective for guiding sequence generation.
3. Through five extensive experiments on three different domains, we demonstrate that CDD achieves state-of-the-art constraint adherence (zero violations across all settings) while preserving high sample quality in terms of fluency, novelty, and coherence.

## 2 Related Work

Recent efforts in controllable text generation have largely focused on constrained decoding, where the output is dynamically restricted to adhere to predetermined syntactic, semantic, or lexical rules. Approaches in this area modify the decoding process to prune the vocabulary, ensuring that only tokens compliant with a formal grammar are considered [11], or employ external modules to filter outputs when direct access to the model's logits is limited [12]. Other methods have also adapted beam search to encourage the presence or absence of specific words [13, 14]. Although these techniques effectively guide the generation process, they depend on augmented sampling heuristics which encourage, but frequently fail to provide, satisfaction of even simple constraint sets [15]. To address this, other research has implemented mappings from natural language to formal logic allowing verification of factual and logical consistency [16–18] and enforcement of feasibility constraints with external modules [19, 20]. However, these approaches, while effective in their respective domains, often require additional, domain-specific components to ensure that the generated outputs are not only syntactically valid but also adhere to practical or logical requirements.

Other controllable generation approaches rely on energy-based sampling procedures to impose constraints on the outputs. These methods typically define an energy function combining language model likelihood with constraint penalties and leverage algorithms such as Metropolis-Hastings to sample low-energy (i.e., high probability) sequences [21–23]. While such methods are highly effective for encouraging simple properties (e.g., style transfer), they inherit many of the theoretical and practical shortcomings of rejection sampling. In particular, hard constraints are typically enforced only probabilistically, making energy-based approaches less suitable for settings requiring strict constraint satisfaction, as is desirable for the tasks considered in this work.

Finally, several gradient-based sampling frameworks have been proposed to impose constraints on token sequences [24]. Building on this approach, Amini et al. [25] improve generation quality via structured Hamiltonian Monte Carlo. However, these methods lack mechanisms for enforcing hard constraints, and are limited to soft attribute control. Other gradient-based control methods for autoregressive generation, such as PPLM [26], apply conditional steering during decoding to bias generations toward desired attributes; we compare against this methods directly, and show that while it can shift output distributions, it fails to reliably enforce structural or semantic constraints. Guided discrete diffusion methods have also been explored for protein design. For example, [27] applies gradient-based guidance to the denoiser's hidden states during sampling to steer sequences toward desired property objectives. Compared to these gradient-based guidance methods, our method frames each denoising step as a constrained optimization and enforces feasibility via an augmented-Lagrangian projection, presenting different behavior and formal guarantees.

## 3 Preliminaries: Discrete Diffusion Models

While diffusion models were originally developed for continuous data, they have recently been extended to discrete domains such as text, enabling non-autoregressive generation [7, 9, 28, 29]. In contrast to autoregressive models which predict tokens one by one, *discrete diffusion* methods generate entire sequences in parallel by first corrupting sequences through a forward noising process and then iteratively reconstructing them with a learned reverse process. *This is a key enabler recognized by this work, which exploits this modus operandi to impose global constraints while simultaneously maintaining high fidelity.*

Let $\boldsymbol{x}_0 = (\boldsymbol{x}_0^1, \ldots, \boldsymbol{x}_0^L)$ denote an input sequence of size $L$, where each token $\boldsymbol{x}_0^i \in \mathcal{V}$ is represented as a one-hot vector over a vocabulary $\mathcal{V}$ of $N$ distinct tokens. In discrete diffusion models, the forward process produces, at time $t \in [0, T]$, a sequence of values $\boldsymbol{x}_t \in \mathcal{V}^L$ that parametrizes probability distributions for each token $\boldsymbol{x}_t^i$ in the sequence. We denote the corresponding sequence of predicted tokens by $\boldsymbol{x}_t^\star = \arg\max(\boldsymbol{x}_t)$ where the $\arg\max$ operator is applied to every member $\boldsymbol{x}_t^i$ of the sequence $\boldsymbol{x}_t$. The diffusion process specifies a *forward transition*, defined as:

$$q(\boldsymbol{x}_t \mid \boldsymbol{x}_0) = \mathrm{Cat}\big(\boldsymbol{x}_t; \alpha_t \boldsymbol{x}_0 + (1 - \alpha_t)\,\nu\big), \tag{1}$$

where $\alpha_t$ is a schedule that decreases over time, $\mathrm{Cat}(\cdot; p)$ is the categorical distribution over probability vector $p \in \Delta^N$, and $\nu$ is a fixed reference distribution that specifies the type of corruption applied to each token. For example, the *Masked Diffusion Language Model* (MDLM) [7] uses a one-hot vector corresponding to the `[MASK]` token for $\nu$. In contrast, *Uniform Diffusion Language Models* (UDLM) [8], uses $\nu$ as the uniform distribution over the vocabulary. Conversely, this instantiation allows tokens to be re-perturbed in later time steps.

In the *reverse process*, $\boldsymbol{x}_T$ is initialized from $\nu$ (e.g., MDLM initializes the probability vectors as one-hots where the weight is fully concentrated on the `[MASK]` token). For each position $i$ in the sequence, while $\boldsymbol{x}_t^i = \nu$, the denoiser parameterizes the transitions to clean data, $\boldsymbol{x}_0$, and the probability of $\boldsymbol{x}_t^i = \nu$ is discounted as $t \to 0$. Once $\boldsymbol{x}_t \neq \nu$, the probability vector transitions into a one-hot vector, concentrating all weight exclusively on the predicted token.

$$q(\boldsymbol{x}_s \mid \boldsymbol{x}_t, \boldsymbol{x}_0) = \begin{cases} \mathrm{Cat}\big(\boldsymbol{x}_s; \boldsymbol{x}_t\big), & \text{if } \boldsymbol{x}_t \neq \nu, \\ \mathrm{Cat}\Big(\boldsymbol{x}_s; \frac{(1-\alpha_s)\,\nu + (\alpha_s - \alpha_t)\mathbf{x}_\theta(\boldsymbol{x}_t, t)}{1 - \alpha_t}\Big), & \text{if } \boldsymbol{x}_t = \nu. \end{cases} \tag{2}$$

Equation (2) formalizes this reverse process parameterization, where $\mathbf{x}_\theta(\boldsymbol{x}_t, t)$ denotes the trained denoiser and $s$ is the subsequent timestep in the continuous time reverse process and $0 \leq s < t \leq T$. Despite different corruption kernels (absorbing `[MASK]` [7] vs. uniform [8]), both samplers expose the full token–probability tensor $x_t$ for all positions at every reverse step.

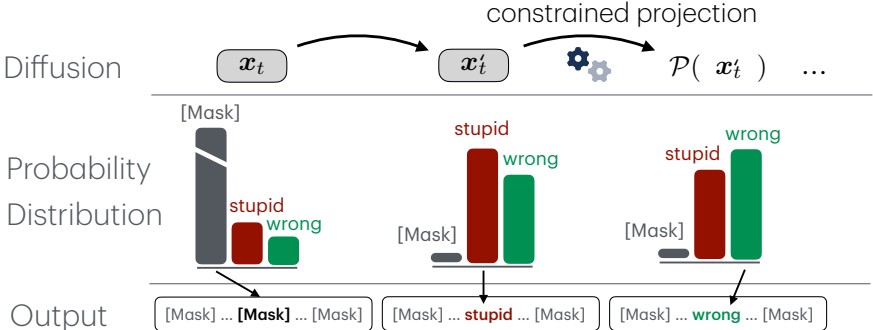

Figure 2: Illustration of CDD's projection step embedded throughout the sampling process.

# 4 Constrained Discrete Diffusion (CDD)

## 4.1 Projected diffusion sampling.

We begin by introducing a perspective of the discrete diffusion reverse process that motivates our approach. Prior work has shown that *continuous* score-based diffusion sampling processes can be framed as an sequential optimization procedure [30]. Score-based parameterizations enable this framing as the model learns to predict the gradients of the data density function, $\mathbf{x}_\theta(\boldsymbol{x}_t, t) \approx \nabla \log q_t(\boldsymbol{x}_t \mid \boldsymbol{x}_0)$ through Score Matching, and these gradients can then be applied to optimize $\boldsymbol{x}_t$ to the data distribution $q_t(\boldsymbol{x}_t)$. This is typically conducted through a Langevin dynamics algorithms [31], which is used either directly [32] or through predictor-corrector Euler-discretizations [33]. In both cases, a series of $M$ steps of Langevin dynamics are taken for a fixed distribution $q_t(\boldsymbol{x}_t)$:

$$\boldsymbol{x}'_{t(\ell)} \xleftarrow{\text{update } (\ell)} \boldsymbol{x}_{t(\ell)} + \gamma_t \nabla_{\boldsymbol{x}_{t(\ell)}} \log q_t(\boldsymbol{x}_{t(\ell)}|\boldsymbol{x}_0) + \sqrt{2\gamma_t}\epsilon \qquad (\text{for } \ell = 1 \dots M), \qquad (3)$$

where $\gamma_t$ is the step size, $\epsilon$ is added noise, and $\log q_t(\boldsymbol{x}_t)$ is the density function of the *learned* distribution at time step $t$. Note that while *annealing* is used to improve convergence, it is applied only after every $M$ iterations. At that point, the sample is updated ($\boldsymbol{x}_{t(M)} \to \boldsymbol{x}_{t-1(1)}$) and the model transitions to the next distribution, $q_{t-1}(\boldsymbol{x}_{t-1} \mid \boldsymbol{x}_0)$, however, the step size and the distribution $q_t(\boldsymbol{x}_t \mid \boldsymbol{x}_0)$ remain stationary throughout these $M$ iterations.

Discrete diffusion models can leverage a discrete generalization of the score function, referred to as the *Concrete score*, to approximate the gradient of the probability density function [7, 9, 34]. Concrete Score Matching provides an approach mirroring continuous Score Matching in that the estimated gradients of the probability density function are used to guide the sample to high density regions of the target distribution. While not always explicitly framed as Concrete Score Matching [9, 34], the denoiser often implicitly models the score function, as supported by theoretical results in [7] demonstrating its equivalence in simplified formulations ($\nabla \log q_t(\boldsymbol{x}_t \mid \boldsymbol{x}_0) \approx \langle \mathbf{x}_\theta(\boldsymbol{x}_t, t), \mathbf{y} \rangle \ \forall \mathbf{y} = 1, \dots, N$, in this case). This enables the use of Langevin-based samplers for discrete diffusion, as commonly employed, e.g., in [34].

Effectively, under some regularity conditions, Langevin dynamics will converge towards a stationary point. As shown by Xu et al., Langevin dynamics acts as an "almost-minimizer" on the optimized function, which in this case will be the negative probability density function, converging within a fixed bound of the optimum. Hence, for each series of steps with a stationary density function, within this fixed bound, the sampling procedure can be framed as the optimization problem,

$$\underset{\boldsymbol{x}_t}{\text{minimize}} \sum_{t=T}^{1} - \log q_t(\boldsymbol{x}_t|\boldsymbol{x}_0) \quad \text{s.t.} \ \ \boldsymbol{x}_t \in \mathbf{C}. \qquad (4)$$

Here, Equation (4) extends the representation from an unconstrained optimization problem to a constrained version by introducing a constraint set $\mathbf{C}$. However, while iteratively applying the denoiser enables sampling from the posterior distribution, it does not natively incorporate externally defined constraints, or even implicit ones, given the stochastic nature of the process. Previous work

for continuous modalities has proposed enforcing $\boldsymbol{x}_t \in \mathbf{C}$ by applying a Euclidean projection after each denoising step [30], which is natural for continuous modalities which fall in a real space, but this is misaligned when applied to discrete diffusion which operates over a probability simplex $\boldsymbol{x}_t \in \Delta^N$.

To address this, we introduce a projection operator that minimizes the *Kullback-Leibler (KL) divergence* between the projected and original probability distributions, as opposed to a Euclidean distance metric. Given the model's predicted probabilities, the projection is defined as:

$$\boldsymbol{x}_s = \boldsymbol{x}_{t(\ell+1)} = \mathcal{P}_{\mathbf{C}}\left(\boldsymbol{x}'_{t(\ell)}\right) \stackrel{\text{def}}{=} \arg\min_{\boldsymbol{y}} D_{\text{KL}}\left(\boldsymbol{x}'_{t(\ell)}\|\boldsymbol{y}\right) \quad \text{s.t. } \arg\max(\boldsymbol{y}) \in \mathbf{C}. \tag{5}$$

The integration of this projection operator ensures that $\boldsymbol{x}_s$ is the "nearest sample" that falls within the feasible subdistribution according to a distance metric defined over probability distributions. This ensures the denoising trajectory remains within the allowable set when $\mathbf{C}$ is convex, enabling effective navigation along its boundary or interior toward an optimal solution (illustrated in Figure 2). Next, we show how this projection operator can be formulated as a subproblem within the sampling procedure and efficiently solved using gradient-based methods. Moreover, while convergence guarantees are available for convex constraint sets, as discussed below, Section 5 demonstrates how this technique can effectively handles highly non-linear constraints, including toxicity mitigation, molecular generation, and instruction following, achieving zero violations across all cases.

Importantly, while state transitions are imposed on the token probability distributions, constraint satisfaction is evaluated on the *decoded sequence*. Indeed, the constraints are formulated such that the $\arg\max$ of the projected sequence $\boldsymbol{y}$ (referred to as $\boldsymbol{y}^\star$), must satisfy sentence level criteria $\boldsymbol{y}^\star \in \mathbf{C}$.[3] However, this use of a non-differentiable $\arg\max$ operation also poses challenges to the resolution of the projection, which relies on gradient-based optimization.

## 4.2 Differentiable projection.

To address the non-differentiability of the $\arg\max$ function, we apply a Gumbel-Softmax relaxation $\tilde{\phi}$, which approximates discrete token probabilities with continuous values [36] as

$$\tilde{\phi}(\boldsymbol{x}_t^i)(v) = \frac{\exp\left(\frac{\log \boldsymbol{x}_t^i(v) + \xi_v}{\text{T}_{\text{sample}}}\right)}{\sum_{v'=1}^N \exp\left(\frac{\log \boldsymbol{x}_t^i(v') + \xi_{v'}}{\text{T}_{\text{sample}}}\right)}, \tag{6}$$

where $\boldsymbol{x}_t^i(v)$ is the probability of token $v$ in the vocabulary, $\xi_v$ is drawn from the Gumbel$(0,1)^1$ distribution for token $v$, and $\text{T}_{\text{sample}} > 0$ is the temperature parameter controlling the smoothness of the output. This enables gradient propagation during the projection step, while closely approximating the discrete $\arg\max$ operation.

## 4.3 Augmented Lagrangian projection.

Consider a generic constraint on a sequence $\boldsymbol{x}$ defined via a measurable property given by a function $g_i(\boldsymbol{x})$. For instance, $g_i(\cdot) : \mathbb{R}^{L \times N} \to \mathbb{R}^+$ might evaluate sentence toxicity (see Section 5.1), count the number of words occurring in a string (see Section 5.3), or even serve as a black-box function computed by an external routine, as in our molecular sequence generation application (see Section 5.2). To guide the projection operations, we require a measure of constraint violation that can later inform the parametrization of our projection update rule; for ease of presentation, we express the constraint in the form $g_i(\boldsymbol{x}) < \tau_i$, where $\tau_i \geq 0$ represents an acceptable threshold that must not be exceeded. To quantify by how much a given sequence violates the constraint, we define

$$\Delta g_i(\tilde{\phi}(\boldsymbol{x}_t)) = \max\left(0, g_i(\tilde{\phi}(\boldsymbol{x}_t)) - \tau_i\right),$$

where $g = (g_1, \ldots, g_m)$ can be treated as series of functions corresponding to a series of thresholds $\tau = (\tau_1, \ldots, \tau_m)$, and $\Delta g = (\Delta g_1, \ldots, \Delta g_m)$ is defined in analogously quantifying $m$ constraints.

In practice, $\Delta g$ is non-linear, and, thus, to implement Equation (5), we adopt an augmented Lagrangian approach [37]. In augmented Lagrangian methods, the problem constraints are incorporated into the objective of a minimizer via Lagrange multipliers $\lambda$ and a quadratic penalty term $\mu$. Let $\boldsymbol{x}'_t$

---

[3]We assume a greedy decoding scheme as is standard to current diffusion-based language models.

be the probability distribution after the denoising step at diffusion time $t$. We introduce a projected distribution $\boldsymbol{y}$, which is iteratively updated to reduce the constraint violations (measured by the score $\Delta g$) while remaining close (in KL-divergence) to $\boldsymbol{x}'_t$. Concretely the augmented Lagrangian dual function is defined as:

$$\mathcal{L}_{\mathrm{ALM}}(\boldsymbol{y}, \boldsymbol{x}'_t; \lambda, \mu) = D_{\mathrm{KL}}\left(\boldsymbol{x}'_t \| \boldsymbol{y}\right) + \sum_{i=1}^{m} \lambda_i \Delta g_i(\tilde{\phi}(\boldsymbol{y})) + \frac{\mu_i}{2} \Delta g_i(\tilde{\phi}(\boldsymbol{y}))^2$$

where $\lambda = (\lambda_1, \ldots, \lambda_m)$ is a non-negative Lagrange multiplier and $\mu = (\mu_1, \ldots, \mu_m)$ is a non-negative quadratic penalty term. When using a Lagrangian function, the primal optimization problem becomes

$$\arg\min_{\boldsymbol{y}} \mathcal{L}_{\mathrm{ALM}}(\boldsymbol{y}, \boldsymbol{x}'_t; \lambda, \mu),$$

and is a lower bound of the original projection operator (5) by weak duality [37]. To obtain the strongest Lagrangian relaxation of the projection, the *Lagrangian dual* can be used to find the best Lagrangian multipliers $\lambda$ and penalty terms $\mu$, i.e.,

$$\arg\max_{\lambda, \mu}\left(\arg\min_{\boldsymbol{y}}\left(\mathcal{L}_{\mathrm{ALM}}(\boldsymbol{y}, \boldsymbol{x}'_t; \lambda, \mu)\right)\right). \tag{7}$$

In practice, the Lagrangian dual is a strong approximation of the original problem (our projection into the constraint space).

The optimization of (7) proceeds iteratively, following a gradient-based update on $\boldsymbol{y}$ while dynamically adjusting the Lagrange multiplier $\lambda$ and penalty coefficient $\mu$. Specifically, we perform the following updates [38]:

$$\boldsymbol{y} \leftarrow \boldsymbol{y} - \eta \nabla_{\boldsymbol{y}} \mathcal{L}_{\mathrm{ALM}}(\boldsymbol{y}, \boldsymbol{x}'_t; \lambda, \mu) \tag{8a}$$

$$\lambda \leftarrow \lambda + \mu \Delta g(\boldsymbol{y}^\star), \tag{8b}$$

$$\mu \leftarrow \min(\alpha\mu, \mu_{\max}), \tag{8c}$$

where $\eta$ is the gradient step size, $\alpha > 1$ is a scaling factor that progressively increases $\mu$ over iterations, and $\mu_{\max}$ is an upper bound on the penalty term. These updates drive $\boldsymbol{y}$ as close as possible to satisfy $\Delta g(\tilde{\phi}(\boldsymbol{y}^\star)) \leq \tau$ while also ensuring it remains close to the original denoised distribution $\boldsymbol{x}'_t$. Pseudocode is provided for our implementation in Algorithm 1.

---

**Algorithm 1:** Augmented Lagrangian

**Init:** $\lambda, \mu, \eta, \alpha, \delta$;
**Input:** probability distribution $\boldsymbol{x}'_t$
$\boldsymbol{y} \leftarrow \boldsymbol{x}'_t$;
**while** $\Delta g(\boldsymbol{y}^*) < \delta$ **do**
    **for** $j \leftarrow 1$ **to** `max_inner_iter` **do**
        $\mathcal{L}_{KL} \leftarrow D_{\mathrm{KL}}(\boldsymbol{x}'_t \| \boldsymbol{y})$
        $\mathcal{L}_{\mathrm{viol}} \leftarrow \sum_{i=1}^{m} \lambda_i \Delta g_i(\tilde{\phi}(\boldsymbol{y})) +$
        $\frac{\mu_i}{2} \Delta g_i(\tilde{\phi}(\boldsymbol{y}))^2$
        $\mathcal{L}_{\mathrm{ALM}} \leftarrow \mathcal{L}_{KL} + \mathcal{L}_{\mathrm{viol}}$
        $\boldsymbol{y} \leftarrow \boldsymbol{y} - \eta \nabla_{\boldsymbol{y}} \mathcal{L}_{\mathrm{ALM}}$
    $\lambda \leftarrow \lambda + \mu \Delta g(\boldsymbol{y}^*)$
    $\mu \leftarrow \min(\alpha\mu, \mu_{\max})$
$\boldsymbol{x}_s \leftarrow \boldsymbol{y}$;
**Output:** $\boldsymbol{x}_s$

---

To assess the robustness of this projection formulation, we performed an ablation study (Appendix C) examining the sensitivity of the Lagrangian hyper-parameters. Across the full grid of settings, *the Lagrangian relaxation converges to feasible solutions in all our test cases.*

As shown in the next section, there is a high degree of flexibility in how these constraints can be implemented. For instance, they can be implemented as surrogate models (e.g., a classifier) that can be used to provide a continuous score, allowing for smooth gradient-base updates, as shown above.

## 4.4 Theoretical justification.

The next result shows that the constrained reverse diffusion process converges to samples within the feasible region $\boldsymbol{C}$ while also keeping their distribution close to the data manifold, thus ensuring that the generated samples are both valid and realistic. Let $D_{\mathrm{KL}}(\boldsymbol{x}_t, \boldsymbol{C}) = \inf_{\boldsymbol{y} \in \boldsymbol{C}} D_{\mathrm{KL}}(\boldsymbol{y} \| \boldsymbol{x}_t)$ denote the KL divergence from $\boldsymbol{x}_t$ to the set $\boldsymbol{C}$.

**Theorem 4.1** (Convergence of CDD). *Let $\boldsymbol{C}$ be non-empty and $\beta$-prox-regular in the sense of [39, Def. 13.27], and the score network satisfy $\|\nabla_{\boldsymbol{x}_t} \log q_t(\boldsymbol{x}_t)\| \leq G$ (a standard consequence of the bounded-data domain after normalization). Then, for positive step sizes $\gamma_t, \leq \frac{1}{2G^2}\beta$, the following inequality holds for the distance to the feasible set $\boldsymbol{C}$:*

$$D_{\mathrm{KL}}(\boldsymbol{x}'_s, \boldsymbol{C}) \leq (1 - \boldsymbol{\alpha}_t) D_{\mathrm{KL}}(\boldsymbol{x}'_t, \boldsymbol{C}) + \boldsymbol{\alpha}_{t+1}^2 G^2, \qquad \text{(non-asymptotic feasibility)}$$

*where $\boldsymbol{\alpha}_t$ is proportional to the discrete Langevin step size $\gamma_t$ and $G$ bounds the score norm.*

| | | Toxicity | | | | | | |
|---|---|---|---|---|---|---|---|---|
| **Model** | **Size** | **PPL** | | **LLM-Judge** | **Coherence** | **Viol (%)** | | |
| | | Mean | Median | Coherence | Change (%) | $\tau=0.25$ | $\tau=0.50$ | $\tau=0.75$ |
| GPT2 | 124M | 18.78 | 17.96 | 42.68 | – | 33.2 | 21.6 | 13.1 |
| GPT2$_{\text{PPLM}}$ | 124M | 46.40 | 34.05 | 18.88 | -55.8 | 16.1 | 8.4 | 4.0 |
| GPT2$_{\text{FUDGE}_{\lambda=2}}$ | 124M | 26.46 | 18.79 | 41.31 | -2.1 | 31.5 | 19.7 | 12.7 |
| GPT2$_{\text{FUDGE}_{\lambda=9}}$ | 124M | 81.84 | 19.22 | 38.90 | -8.9 | 30.6 | 19.6 | 11.7 |
| Llama 3.2 | 1B | **15.66** | **14.58** | **57.10** | – | 34.9 | 27.8 | 23.1 |
| MDLM | 110M | 46.72 | 39.75 | 20.02 | – | 32.1 | 23.2 | 17.2 |
| **CDD**$_{\tau=0.25}$ **(Ours)** | 110M | 61.55 | 45.42 | 20.16 | +0.7 | **0.0** | **0.0** | **0.0** |
| **CDD**$_{\tau=0.50}$ **(Ours)** | 110M | 59.44 | 44.24 | 20.30 | +1.4 | – | **0.0** | **0.0** |
| **CDD**$_{\tau=0.75}$ **(Ours)** | 110M | 54.87 | 43.51 | 20.88 | **+4.3** | – | – | **0.0** |

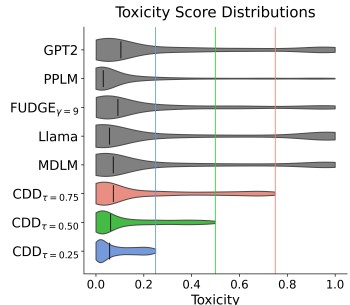

Figure 3: Results across different toxicity thresholds, PPL percentiles, LLM-Judge coherence, and coherence degradation levels. PPL is evaluated with GPT-2-XL, and coherence metrics are assessed by the LLM-Judge. **Bold** and underlined values denote the best and second-best, respectively.

Theorem 4.1 shows that the distance to the feasible set $C$ decreases at a rate of $1 - \boldsymbol{\alpha}_t$ at each step (up to an additive $\boldsymbol{\alpha}_t^2 G^2$ noise). This implies $\varepsilon$-feasibility after $\mathcal{O}(\boldsymbol{\alpha}_{\min}^{-1})$ steps, with $\boldsymbol{\alpha}_{\min} \propto \min_t \gamma_t$, and a cumulative KL drift within $\mathcal{O}(\sum_t \boldsymbol{\alpha}_t)$. The theorem assumes $\beta$-prox-regularity, which provides a relaxation of typical convexity assumptions, and implies that for each viable point and normal direction, small perturbations still project uniquely and smoothly back to $C$. Theorem proofs are provided in Appendix F.

# 5 Experiments

To empirically demonstrate the advantage provided by CDD, we compare our method against similarly sized autoregressive and diffusion-based language models. Specifically, we benchmark our performance on constrained text generation against autoregressive baselines: GPT-2 and Llama 3.2 [40, 41], and discrete diffusion baselines MDLM and UDLM [7, 8]. We intentionally select models of similar size to, and up to one magnitude larger than, our base discrete diffusion model, noting that we expect the performance of both classes of models to scale similarly with size. As demonstrated by Schiff et al. [8], MDLM outperforms UDLM on the natural language tasks discussed in Sections 5.1 and 5.3. Consequently, we compare to both models in Section 5.2 and solely MDLM for the following natural language tasks. We use UDLM as the base discrete diffusion model for Section 5.2, and MDLM in Sections 5.1 and 5.3. For each application, CDD uses configurations as described in [7, 8] unless otherwise specified. Additional experiment details are available in Appendix B.

## 5.1 Natural Language Toxicity Mitigation

While LLMs have demonstrated remarkable capabilities in generating human like text, a byproduct of data driven generation is the inadvertent creation of harmful, offensive and dangerous text. Despite undergoing alignment finetuning, current state-of-the-art models still frequently generate harmful outputs. While post-processing methods can be effective in moderating generations, they can be circumvented by adversarial or malicious prompts, resulting in toxic output generation. In this experiment, we generate outputs conditioned on prompts from the RealToxicityPrompts dataset [42]. We select prefixes with toxicity score above 0.5, testing how these models perform on toxic inputs, and perplexity below 30.0, avoiding incoherent generations. To model the constraints, GPT-Neo [43] is finetuned on the Jigsaw Toxic Comment Classification Challenge[4]. We evaluate on 1,000 samples, and discuss ALM hyper-parameters in Appendix C, runtime in Appendix D, and diversity as measured by entropy [10] in Appendix E.

Figure 3 presents a comprehensive evaluation on model performance for toxicity mitigation, where the objective is to have the generated output satisfy some toxicity threshold, based on a surrogate model, while maintaining competitive perplexity and coherence. We report perplexity (PPL), coherence as determined by a LLM judge, and violation rates at toxicity thresholds $\tau = 0.25$, $\tau = 0.50$, and $\tau = 0.75$ across different model architecures and baselines.

---

[4]https://www.kaggle.com/c/jigsaw-toxic-comment- classification-challenge

Among the baselines, GPT2 (124M) and Llama (1B) achieve the lowest perplexity and highest coherence score, however, they frequently produce toxicity content resulting in high violations. Although PPLM and FUDGE (124M) [15, 26] are able to decrease the violations for all toxicity thresholds, they are unable to consistently prevent toxic generations while suffering from an increase in perplexity and a decrease in coherence. MDLM (110M) exhibits higher perplexity than GPT2 and lower coherence, and similarly shares a non-negligible amount of toxicity constraint violations. In contrast, CDD provides **perfect constraint satisfaction** at each toxicity thresholds, maintaining comparable perplexity scores to its base diffusion language model and other baselines. Additionally, while constraint imposition degrades the coherence of the controllable generation baselines, CDD observes an increase in coherence as evaluated by the LLM-Judge. *These findings are important as they show how, using projection-based control strategies, it is possible to achieve complete toxicity constraint satisfaction at desired thresholds.*

## 5.2 Molecular Generation

Next, we show the ability of CDD to impose constraints on a scientific task, specifically in the domain of molecular generation. In this experiment we generate SMILES strings [44], a linear representation of a molecule's structure with a vocabulary consisting of a limited vocabulary of forty symbols and a grammar dictating molecular validity. Recent advances in generative modeling have enabled the design of molecular sequences by leveraging techniques from natural language processing. Despite their impressive ability to optimize for specific chemical properties, these models often generate molecules that fall short of practical requirements, either by producing compounds that are difficult to synthesize or by closely mimicking existing structures. Furthermore, generated molecules often fail to maximize qualitative properties such as drug-likeness (QED) [45] that are critical to the practicality of the generations and used as a qualitative metric for the experiments discussed in this section.

To address this challenge, our framework incorporates two distinct constraint mechanisms: **(1)** *synthetic feasibility*, which ensures that generated molecules can be synthesized in a laboratory setting [46], and **(2)** *novelty*, which guarantees that the generated molecules are not already present in training datasets. For each constraint, we implement a dedicated projection operator, described in B, that imposes hard limits on the generation process, resulting in final outputs that adhere precisely to our desired standards of synthetic accessibility or novelty.

**Synthetic accessibility constraints.** Synthetic accessibility is commonly assessed using black-box, non-differentiable functions. To enable gradient-based optimization, we train a surrogate model on the QM9 dataset [47], where training labels are derived from RDKit's synthetic accessibility score [48]. This allows us to approximate a differentiable function $g(\cdot)$ for synthetic accessibility. While our molecular generation framework employs this surrogate model to optimize for synthetic accessibility during training, the actual assessment of accessibility violations is conducted by a separate, black-box external model. This external evaluation rigorously measures the degree to which generated molecules comply with the synthetic accessibility criteria, using a series of thresholds ($\tau = 3.0, 3.5, 4.0,$ and $4.5$). For the discrete diffusion baselines, we adapt the guidance mechanisms provided by [8] for QED by retraining on synthetic accessibility labels for Classifier-Based Guidance (CBG) and Classifier-Free Guidance (CFG). A similar adaptation is applied for the autoregressive baselines employing FUDGE guidance [15]. The results, summarized in Figure 4 (left), confirm that CDD achieves perfect compliance across all thresholds. This 0% violation rate is achieved, all while generating a competitive number of valid molecules and exhibiting the highest drug-likeness scores.

**Novelty constraints.** Novelty in molecule generation refers to the model's ability to produce new chemical structures that were not explicitly contained in its training set. In the context of generative modeling for drug design or other chemistry applications, novelty is often an important objective for the reasons of chemical space exploration and practical relevance. In this experiment, we generate SMILES strings constraining all valid generations to be novel. The novelty constraint is enforced at every denoising step. If the current candidate sequence already exists in an external database, a projection operator minimally perturbs its token-probability vector to yield an unseen molecule, thereby approximating a gradient step through the otherwise non-differentiable novelty indicator. Specifically, the novelty projection operator is a best-first traversal of the token-probability space: each token flip incurs a cost equal to its probability gap from the argmax, a priority queue retrieves the lowest-cost unseen sequence, and we then renormalize and cache it to prevent duplication. As it selects the sequence with the minimal cumulative flip cost, our distance metric over sequences, the

| Molecules (Synthetic Accessibility) | | | | | | | |
|---|---|---|---|---|---|---|---|
| Model | Valid | Novel | QED | Viol (%) | | | |
| | | | | $\tau = 3.0$ | $\tau = 3.5$ | $\tau = 4.0$ | $\tau = 4.5$ |
| AR | 1023 | 0 | 0.46 | 91.6 | 78.4 | 62.3 | 42.0 |
| AR$_{\text{FUDGE } \gamma = 7}$ | 925 | 6 | 0.48 | 11.1 | 9.8 | 9.6 | 9.2 |
| MDLM | 596 | 20 | 0.45 | 85.9 | 73.7 | 61.1 | 44.0 |
| MDLM$_{\text{D-CFG } \gamma = 3}$ | 772 | 3 | 0.41 | 87.8 | 73.9 | 54.2 | 22.5 |
| MDLM$_{\text{D-CBG } \gamma = 3}$ | 436 | 14 | 0.37 | 50.5 | 48.6 | 46.1 | 44.7 |
| UDLM | 895 | 21 | 0.47 | 89.4 | 88.0 | 58.1 | 37.8 |
| UDLM$_{\text{D-CFG } \gamma = 5}$ | 850 | 18 | 0.47 | 80.6 | 58.6 | 35.9 | 13.9 |
| UDLM$_{\text{D-CBG } \gamma = 10}$ | 896 | 28 | 0.47 | 90.1 | 77.8 | 58.6 | 37.7 |
| **CDD$_{\tau = 3.0}$(Ours)** | 353 | **36** | **0.63** | **0.0** | **0.0** | **0.0** | **0.0** |
| **CDD$_{\tau = 3.5}$(Ours)** | 863 | 22 | 0.62 | – | **0.0** | **0.0** | **0.0** |
| **CDD$_{\tau = 4.0}$(Ours)** | 936 | 31 | 0.61 | – | – | **0.0** | **0.0** |
| **CDD$_{\tau = 4.5}$(Ours)** | 938 | 33 | 0.58 | – | – | – | **0.0** |

| Molecules (Novelty) | | | | |
|---|---|---|---|---|
| Model | Size | Valid & Novel | QED | Viol (%) |
| No Guidance | | | | |
| AR | 92M | 11 | 0.41 | 98.93 |
| MDLM | 92M | 271 | 0.45 | 54.53 |
| UDLM | 92M | 345 | **0.46** | 61.45 |
| **CDD (Ours)** | 92M | **511** | 0.45 | **0.00** |
| CFG | | | | |
| AR$_{\text{D-CFG } \gamma = 3}$[†] | 92M | 79 | 0.60 | 91.61 |
| MDLM$_{\text{D-CFG } \gamma = 3}$[†] | 92M | 96 | 0.60 | 69.82 |
| UDLM$_{\text{D-CFG } \gamma = 5}$[†] | 92M | 64 | **0.62** | 93.69 |
| **CDD$_{\text{D-CFG } \gamma = 5}$ (Ours)** | 92M | **251** | 0.60 | **0.00** |
| CBG | | | | |
| AR$_{\text{FUDGE } \gamma = 7}$[†] | 92M | 53 | **0.61** | 94.28 |
| MDLM$_{\text{D-CBG } \gamma = 3}$[†] | 92M | 117 | 0.58 | 72.08 |
| UDLM$_{\text{D-CBG } \gamma = 10}$[†] | 92M | 64 | **0.61** | 93.59 |
| **CDD$_{\text{D-CBG } \gamma = 10}$ (Ours)** | 92M | **355** | 0.59 | **0.00** |

Figure 4: **Left:** Results for synthetic accessibility constrained molecule generation constraints. QED and constraint violations are reported for only valid molecules, and novel molecules must be valid and have no violation ($\tau \leq 3.0$). **Right:** Results for novelty projection with and without QED guidance. Violation represents percentage of valid, but not novel molecule generations. QED is reported for only novel molecules. Results denoted with [†] are as reported by Schiff et al. [8]. **Bold** and underlined values mark the best and second-best, respectively.

procedure is mathematically equivalent to projecting the distribution onto the set of novel sequences. Repeating this operation throughout the diffusion trajectory guarantees that only new compounds are emitted while preserving the exploratory capacity of the discrete diffusion model.

We compare against baselines with no QED guidance alongside CFG and CBG QED guidance [8] to evaluate the impact of our novelty constraint. The results are shown in the right side of Figure 4. Notice how the CBG setting yields a 203.4% increase in novel molecule generation, while the CFG setting shows a 161.4% increase. Even without guidance, the method still produces 48.1% more novel molecules, with only a minimal reduction in the QED score. *These results are significant because they demonstrate that CDD can effectively generate novel molecules while maintaining a high QED score, which is crucial for many scientific tasks.*

## 5.3 Instruction Following

Finally, the last application focuses on satisfying user-specified symbolic constraints, a task that many autoregressive models struggle with. We evaluate CDD against such models using two types of constraints. First, we assess the ability of the tested models to accurately answer how many letters are in a word, such as 'How many R's are in *Strawberry*?'. We additionally test lexical constraints based of the COLLIE dataset [49], prompting the models to generate word-indexed constrained sentences, such as 'Generate a sentence where the third-to-last word is *mankind*'. For each constraint, the projection operator to imposes *hard constraints* on the generation, pushing the samples back onto the constraint set.

| | Model | Size | PPL | LLM–Judge | Viol (%) |
|---|---|---|---|---|---|
| Counting | GPT-2 (zero-shot) | 124M | 19.2 ± 0.8 | 57.8 | 100 |
| | GPT-2 (instruction-tuned) | 124M | 23.4 ± 1.3 | 70.9 | 5.0 |
| | Llama 3.2 (zero-shot) | 1B | 34.3 ± 3.8 | 79.5 | 100 |
| | Llama 3.2 (instruction-tuned) | 1B | 30.7 ± 2.9 | **91.8** | 9.1 |
| | MDLM | 169M | **16.9 ± 1.0** | 61.2 | 54.5 |
| | **CDD (Ours)** | 169M | 17.9 ± 1.1 | 60.4 | **0.0** |
| Lexical | GPT-2 (zero-shot) | 124M | 26.6 ± 4.2 | 28.1 | 97.6 |
| | GPT-2 (instruction-tuned) | 124M | 62.2 ± 2.9 | 15.4 | 96.4 |
| | Llama 3.2 (zero-shot) | 1B | **11.3 ± 2.0** | 30.7 | 97.2 |
| | Llama 3.2 (instruction-tuned) | 1B | 66.5 ± 6.1 | 22.9 | 93.6 |
| | MDLM | 169M | 21.9 ± 2.0 | 34.8 | 97.5 |
| | **CDD (Ours)** | 169M | 26.6 ± 4.1 | **65.1** | **0.0** |

Table 1: PPL is assessed using GPT2. **Bold** and underlined mark the best and second-best, respectively.

**Counting constraints.** As illustrated in the top of Table 1, unconstrained GPT-2 and Llama 3.2 fail to satisfy any letter-count requirement. Instruction tuning helps, reducing violation rates to 5.0% (GPT-2) and 9.1% (Llama 3.2), but still leaves a notable error margin. Alternatively, MDLM achieves the lowest perplexity and second-best coherence, but over half of its outputs violate the constraint. By contrast, *CDD fully eliminates violations* while preserving near-best fluency (17.9 perplexity), demonstrating its provision of strict compliance without sacrificing language quality.

**Lexical constraints.** The results for lexical constraints are shown in the bottom of Table 1. The zero-shot and instruction-tuned versions of GPT-2 and Llama 3.2 show similarly high violation rates (about 97%), and instruction tuning only minimally improves these figures. MDLM again delivers relatively high fluency (21.9 perplexity, 34.8 coherence) but fails to address the precise positional requirements (97.5% violations). In stark contrast, *CDD ensures perfect adherence* and the best LLM-as-a-Judge score (65.1 coherence), with only a moderate increase in perplexity, thus confirming the effectiveness of its projection mechanism for enforcing hard lexical constraints. *Taken together, these experiments reveal a fundamental limitation of conventional and instruction-tuned language models in scenarios requiring strict constraints. Conversely, discrete diffusion models, when equipped with constraints, can effectively produces outputs that satisfy both counting and lexical requirements, opening new avenues for applications in controlled text generation.*

We include more evaluation and experiments in the Appendix. Specifically, Appendix B details the experimental setup, Appendix C evaluates CDD's robustness through an extensive hyper-parameter ablation, and Appendix D reports runtime comparisons with the baselines.

## 6 Limitations

There are two key limitations that are currently faced in the development of constrained discrete models. The first regards the size of currently available discrete diffusion models. While current discrete diffusion models provide comparable performance in fluency and perplexity to similarly sized autoregressive models, including state-of-the-art models such as Llama 3.2 1B, their generative capabilities fall short of large-scale language models. Larger scale, closed-source discrete diffusion models have been developed and boasted impressive performance [29], but larger open-source models are yet to be provided to the research community. It is expected that the release of such larger models will further boost the performance of our proposed methods.

The second key limitation concerns an inherent byproduct of constraint imposition: increased computational overhead. While unconstrained discrete diffusion language models have been shown to provide significant speed-ups over autoregressive language models due to the parallelization of token generation [7, 9, 50], the inclusion of constraints within the generation process may limit this benefit. Nevertheless, this trade-off is often justified, as imposing constraints is essential in scenarios where correctness outweighs generation speed, such as ensuring safety-critical outputs or supporting rigorous scientific exploration. In such contexts, achieving accurate and reliable outcomes is prioritized over computational efficiency.

## 7 Conclusion

In this paper we presented Constrained Discrete Diffusion (CDD), a novel method for imposing both structured and unstructured constraints during sequence generation. CDD relies on the adoption of discrete diffusion models which enable the consideration of entire sequences when imposing constraints during the generation process. Offering a key departure from current paradigms for controllable text generation, CDD integrates differentiable optimization methods into the diffusion reverse process. This integration, governed by a rigorous mathematical optimization framework, provides precise control for discrete sequence generation. The method is empirically validated on (i) toxicity mitigation and harmful content prevention, (ii) generation of novel molecular sequences that adhere to specific properties for drug discovery, and (iii) imposition of lexical constraints at character and sequence levels, reporting state-of-the-art results for controllable generation in both terms of constraint adherence and quality metrics.

# Acknowledgments

This material is based upon work supported by the National Science Foundation Graduate Research Fellowship Program under Grant No 2234693. Any opinions, findings, and conclusions or recommendations expressed in this material are those of the authors and do not necessarily reflect the views of the National Science Foundation. This work was partially supported by NSF grants CAREER-2401285, RI-2533631, RI-2334936, DARPA under Contract No. #HR0011252E005, and UVA's National Security Data & Policy Institute, ODNI Contracting Activity #2024-24070100001 The work has also been supported by LLNL under Contract DE-AC52- 07NA27344 and by the LLNL-LDRD Program under Project No. 24-ERD-010 and 24-SI-008. This manuscript has been co-authored by Lawrence Livermore National Security, LLC under Contract No. DE-AC52-07NA27344 with the U.S. Department of Energy. Any opinions, findings, and conclusions or recommendations expressed in this material are those of the authors and do not necessarily reflect the views of the NSF, DOE, DARPA, or DOD.

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

# A   Expanded Related Work

**Constrained Decoding Algorithms.**   A prominent strategy in controllable text generation is to employ constrained decoding algorithms, which incrementally parse outputs under specified formatting or structure requirements. Such approaches have been extensively used in tasks like code generation [51], semantic parsing [52], SQL queries [53], and grammar-constrained text generation [11]. These methods typically prune the vocabulary at each decoding step to only those tokens that comply with the target representation's syntax or semantics. For instance, Geng et al. confine the logit distribution to tokens valid within a formal grammar, then employ a *completion engine* to extend partial sequences while maintaining grammatical consistency. Meanwhile, approaches that treat the original language model as a black box resort to external constraints. For example, [12] harnesses a secondary model to parse and restrict outputs, an option that becomes necessary when direct access to the logits is unavailable. However, these workarounds may suffer performance penalties due to their indirect control over the primary model's probability distribution.

Beyond syntax-focused constraints, lexical constraints have also been extensively studied. Lu et al. [13] devise Neurologic decoding, which encourages the presence or absence of specific words in generated text via a modified beam search. They later improve adherence to these constraints by employing an A* search variant [14]. Similar frameworks combine toxicity classifiers with beam search to steer generation toward more benign text [26, 54]. Although these methods can effectively address a range of constraints, they generally rely on augmented sampling heuristics or additional classifiers rather than natively integrating constraints into the generation process.

**Natural Language Verification with Formal Logic.**   Another line of work has examined mapping natural language into logical representations to ensure factual and logical consistency [16–18]. By encoding linguistic inputs in a formal system, these methods can detect inconsistencies or factual errors and thereby provide a measure of control over generated outputs. For instance, Olausson et al., Jiang et al., and Zhang et al. convert natural language statements into first-order logic (FOL) and leverage theorem proving to verify factual correctness. However, FOL struggles to capture complex temporal or sequential dependencies, limiting its applicability to planning or process-oriented tasks. Similarly, other work uses entailment models to verify that a conclusion logically follows from a set of premises [58], though these methods generally stop short of actively generating text that satisfies domain-specific constraints.

**Agent Constraint Satisfaction.**   In robotics and embodied AI, natural language planning frequently incorporates specialized external modules to handle feasibility constraints. For example, Chen et al. [19] and Lin et al. [20] defer to domain-specific motion planners to generate viable trajectories, embedding constraint satisfaction directly into the underlying optimization routines. Although these solutions can integrate strong, domain-aware constraints, they do so by delegating control to highly specialized components. On the discrete side, Yang et al. [59] achieves constraint satisfaction by running a validation module at each step of plan construction and discarding infeasible actions through rejection sampling. While effective in certain settings, such methods typically require domain-specific expertise and do not uniformly integrate constraint enforcement into the generative model itself.

# B   Experimental Setup

## B.1   Toxicity Mitigation

**LLM-as-a-Judge Setup.**   To quantify the *fluency* and *coherence* of generated continuations we follow the increasingly adopted LLM-as-a-Judge evaluation. This method has been shown to result in similar evaluation as human judgment [60]. Specifically, each post-prefix sentence is scored by the instruction-tuned `GEMMA-3-27B-IT` model and issued the following prompt:

> *"You are a language expert evaluating the fluency and coherence of the following AI-generated sentence... Give a single integer score between 0 (poor) and 100 (excellent), with no explanation or comments."*

The model returns a single scalar $s \in [0, 100]$ for each sentence. Using an LLM evaluator offers two key advantages: (i) strong correlation with human judgments at negligible annotation cost, and (ii) scalability for thousands of evaluations performed in various experiments.

**Surrogate Model.** As the surrogate model for toxicity task, we use a GPT-Neo (1.3B) model, adapted for binary classification. We finetune this model on the Jigsaw toxicity dataset which includes multiple toxicity-related labels such as toxic, severe toxic, insult, etc. We consolidate these columns into a single binary target (toxic vs. non-toxic).

**PPLM Implementation.** As a baseline for toxicity reduction in natural language generation we include the PPLM methodology [26] . PPLM uses a frozen pretrained GPT-2 generator for output steering. At every decoding step the algorithm runs the usual forward pass, takes a few gradient-ascent steps on the cached key/value activations to maximize the score of a small attribute model which in this case is a toxicity classifier trained on the Jigsaw toxicity dataset. It simultaneously keeps the perturbed logits close to the original distribution with a KL penalty and a geometric-mean fusion term. We use the authors implementation and code structure without architectural modifications. The underlying steering mechanism, loss terms and hyper-parameter values are unchanged from the authors' baseline.

**FUDGE Implementation.** For another toxicity control baseline we additionally implemented FUDGE [15] on the Jigsaw Toxicity dataset. Following the authors implementation we use an LSTM followed by a linear output layer trained to predict future toxicity from partial prefixes. At inference time we add its scaled log non-toxicity probability to GPT-2's next-token logits for the top candidates then sample from the re-weighted distribution. While the original implementation of FUDGE fixes the guidance strength $\lambda = 1$, we perform a hyperparameter seach and sample from the resulting distribution with $\lambda \in \{1, \ldots, 9\}$, displaying the toxicity and perplexity results in Table 2. We use the authors' original code structure, and base language model.

| $\lambda$ | PPL (Avg) | Coherence | Viol % $\tau = 0.25$ | Viol % $\tau = 0.5$ | Viol % $\tau = 0.75$ |
|---|---|---|---|---|---|
| 1 | 20.34 | 41.64 | 33.5 | 20.0 | 13.1 |
| 2 | 26.46 | 41.79 | 31.6 | 19.7 | 12.7 |
| 3 | 36.36 | 40.07 | 33.1 | 20.7 | 13.3 |
| 4 | 43.80 | 40.37 | 31.8 | 21.5 | 13.3 |
| 5 | 54.37 | 40.12 | 31.7 | 21.6 | 13.0 |
| 6 | 65.23 | 39.33 | 33.1 | 21.7 | 13.1 |
| 7 | 75.23 | 38.76 | 32.5 | 20.3 | 12.9 |
| 8 | 76.51 | 39.09 | 32.2 | 20.8 | 12.6 |
| 9 | 81.84 | 38.91 | 30.6 | 19.6 | 11.4 |

Table 2: FUDGE perplexity, coherence, and violation rates at three thresholds for varying $\lambda$.

**Diffusion Guidance Implementation.** While recent literature [8] has proposed adaptations of classifier-based guidance [61] and classifier-free guidance [62] for discrete diffusion models, these approaches have only been employed for settings with limited vocabulary sizes (e.g., molecule generation where the vocabulary is composed of 40 tokens). For natural language tasks which have much larger vocabulary sizes, for instance, the GPT2 tokenizer is composed of over 50,000 tokens– these techniques have not been applied. Currently, this is a scalability issue, as these approaches require that guidance vectors are computed of size $\mathbb{R}^{L \times N}$, making it much simpler and efficient when operating on a limited vocabulary size. Hence, as there is no prior literature to the extent of our knowledge that has scaled these techniques to natural language tasks, we defer comparison to these methods for the molecular generation setting.

**Implementation Details.** For the ALM projection, we initialize using the following hyperparameters: $\lambda_{\text{init}} = 0.0, \mu_{\text{init}} = 1.0, \text{outer\_iter}_{\text{max}} = 1000, \text{inner\_iter}_{\text{max}} = 10, \eta = 0.20$. We choose these setting for the toxicity experiment as they result in the lowest perplexity in the Appendix C ablation study. For natural language toxicity reduction details we use a generation length of 100 tokens.

## B.2  Molecular Generation

**Training Dataset.** For our molecule generation experiments, we utilize the QM9 dataset [47], as used by Schiff et al. [8]. This dataset comprises approximately 133,885 small organic molecules,

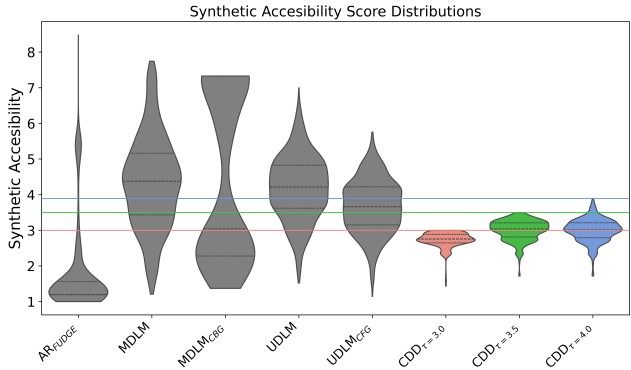

Figure 5: Synthetic Accessibility score distributions for CDD versus competing baselines. Importantly, CDD is the only model that never violates the specified toxicity thresholds.

each encoded as a SMILES string [44], which compactly represents the molecular structure through a sequence of characters. The SMILES representation facilitates discrete modeling by enabling the treatment of molecular generation as a sequence prediction task, making it particularly amenable to discrete diffusion approaches. By leveraging QM9, we can rigorously evaluate the performance of our generative models on tasks that require both the preservation of chemical validity and the precise control of molecular properties, aligning with established protocols in the literature.

### B.2.1 Synthetic Accessibility Constraints

**Surrogate Model.** We train our surrogate model on the QM9 dataset [47] but manually label the training data with RDKIT's synthetic accessibility score [48]. We finetune GPT2 (124M) to act as this surrogate model and directly output a score $s \in [0, 10]$.

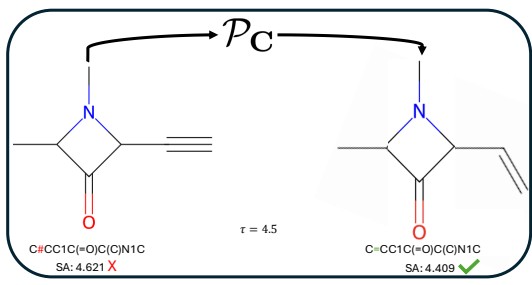

Figure 6: SA projection illustration.

**FUDGE Implementation.** For this setting, we follow the FUDGE implementation provided by [8]. However, while their guidance is trained on QED scores, labeling samples from the QM9 dataset with these scores, we adapt it to label the training data with RDKIT's computation of the synthetic accessibility scores.

**Diffusion Guidance Implementation.** Similar to the implementation of FUDGE, we adapt the guidance mechanisms provided by [8] for QED by retraining on synthetic accessibility labels. Otherwise, our implementation mirrors their approach for QED guidance.

**ALM Implmenetation.** For the ALM projection, we initialize using the following hyperparameters: $\lambda_{\text{init}} = 0.0, \mu_{\text{init}} = 1.0, \mu_{\text{max}} = 1000, \text{outer\_iter}_{\text{max}} = 1000, \text{inner\_iter}_{\text{max}} = 100, \eta = 1.0$.

### B.2.2 Novelty Constraints

**Novelty Projection Operator.** In order to project the molecule sequences into a novel set, we apply a best-first search (BFS) at the probability distribution level. We begin by defining a flip cost for forcing a token to become the new argmax; this cost is the difference between the current top token's probability and the candidate token's probability, or zero if the candidate is already top-ranked. To find a novel sequence at minimal total flip cost, we start with an empty sequence and expand it position by position, accumulating flip costs in a priority queue. Once a full sequence is constructed, we decode it and check if it is absent from our existing dataset. If it is indeed novel, we finalize the sequence, shift the probability distribution so that each selected token becomes the definitive argmax, and insert the resulting sequence into the dataset to prevent re-generation. This procedure systematically maintains high-likelihood sequences while avoiding those already present, terminating for each sample as soon as it finds a suitable novel result before proceeding to the next sample.

**FUDGE Implementation.** For this application, we similarly follow the FUDGE implementation and configuration in [8] for QED guidance.

**Diffusion Guidance Implementation.** We use classifier based guidance and classifier free guidance for QED as implemented in [8].

## B.3 Instruction Following

**LLM-as-a-Judge Setup.** To quantify the *fluency* and *coherence* of generated continuations we follow the increasingly adopted LLM-as-a-Judge evaluation. This method has been shown to result in similar evaluation as human judgment [60]. Specifically, each post-prefix sentence is scored by the instruction-tuned `Gemini-2.0-Flash` model and issued the following prompt:

> *"You are a language expert evaluating the fluency and coherence of the following AI-generated sentence... Give a single integer score between 0 (poor) and 100 (excellent), with no explanation or comments."*

The model returns a single scalar $s \in [0, 100]$ for each sentence. Using an LLM evaluator offers two key advantages: (i) strong correlation with human judgments at negligible annotation cost, and (ii) scalability for thousands of evlautions performed in various experiments.

**Discrete Diffusion Base Model.** While in Section 5.1, we directly utilize the pretrained MDLM checkpoint provided by [7], for the experiments in Section 5.3, we pretrain and subsequently instruction-tune a slightly larger MDLM model, adopting the Llama 1 tokenizer. Empirical observations indicated that employing a marginally larger model and vocabulary size was essential for optimal performance on instruction-guided tasks. Consequently, we adopt this enhanced model for all subsequent evaluations in this context, including both the base MDLM experiments and the proposed CDD framework. Our instruction tuning is conducted using the ShareGPT dataset, and all pretraining follows the design choices and hyperparameters provided by [7].

### B.3.1 Counting Constraints

**Instruction-Tuning Dataset.** We manually construct a dataset of letter counting examples in an instruction-tuning format. We use random words and select a random character from these words 80% of the time. The other 20% of the time we select a letter at random, which may or may not be contained in the word, to represent cases where the correct answer is zero.

**Projection Operator.** Our implementation of this projection operator involves integrating an external module to count the number of times a specific letter appears in a given word. First, we leverage the external module to obtain the accurate count, ensuring that the correct numerical value is determined. Then, during output generation, we adjust the probability distribution over the numerical tokens so that the token corresponding to this count is assigned the highest probability, while the probabilities of other numerical tokens are appropriately diminished. This approach guarantees that when the model is queried about the frequency of a letter in a word, it outputs the correct numerical token with the highest weight in the distribution, maintaining consistency and precision in the final answer.

### B.3.2 Lexical Constraints

**Instruction-Tuning Dataset.** We finetune models on a dataset of examples constructed from evaluation of GPT-4 on the COLLIE dataset provided by Yao et al. [49]. As GPT-4 performs well on these examples, we apply knowledge distillation by compiling these generations into an instruction tuning dataset. The dataset can be directly constructed from the logs included in the code base from Yao et al.

**Projection Operator.** Our implementation of the projection operator begins by parsing the decoded string to locate the position corresponding to the $n_{th}$ word. Because token indices do not necessarily match word indices, owing to the nuances of subword tokenization, we perform an alignment procedure that accurately maps word positions to their respective token indices. Once the index in

the sequence corresponding to the $n_{th}$ word is identified, we apply a closed-form gradient update that adjusts the probability distribution over the vocabulary. This update maximizes the probability of the correct token at the specified position while ensuring that the overall distribution remains valid (i.e., all probabilities sum to one).

## C Hyper-parameter Ablation Study

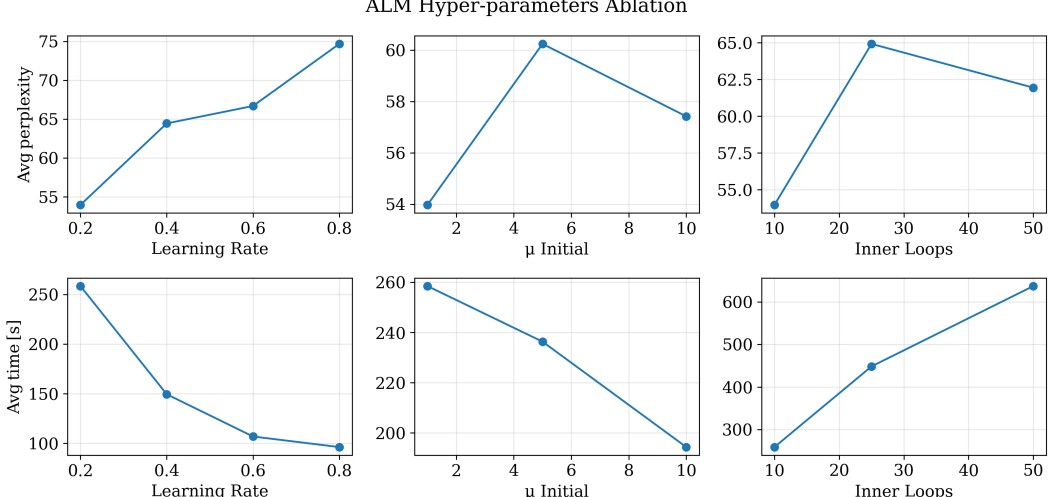

Figure 7: Augmented-Lagrangian Method hyper-parameter ablation study.

To evaluate the robustness of CDD we perform a hyper-parameter sensitivity study for toxicity constraints in natural language generation. We sweep three key Augmented-Lagrangian optimization hyper-parameters: learning-rate, number of inner loop steps, and $\mu$-initialization values. We further examine the impact of projection frequency on constraint generation. In our study we evaluate for constraint satisfaction, perplexity of generated sequences, and runtime. Crucially, for every configuration in the search space the toxicity threshold constraint are **fully satisfied**, indicating that the method remains reliable under substantial hyper-parameter variations. For all test setting we evaluate with the same subset of 200 random samples, and all are run on a single NVIDIA RTX A6000 GPU.

**Augmented-Lagrangian Parameters**  The Augmented-Lagrangian Method lets us express the detoxification goal (*toxicity* $\leq \tau$) as a smooth, differentiable objective rather than a hard discrete constraint. At every outer iteration ALM augments the loss with (i) a linear Lagrange term that guides the solution toward feasibility and (ii) a quadratic penalty that makes constraint violations increasingly costly. This dual penalty keeps gradients informative even when the constraint is initially violated, yet avoids the numerical instability of a single, very large fixed penalty. The inner optimizer therefore steers the logits toward a region that is both close to the original sentence (via the KL term) and certified non-toxic (via the AL penalty). To assess robustness and sensitivity, we perform a grid search over three key ALM hyper-parameters:

1. **Learning rate** $\eta$ controls the size of each optimization step on the logits. Higher $\eta$ accelerates early progress but can overshoot when the penalty weight $\mu$ grows; lower $\eta$ stabilizes training at the cost of more iterations.

2. **Initial penalty weight** $\mu_0$ sets the starting strength of the quadratic term. A small $\mu_0$ allows the optimizer to explore a broader region and preserve fluency, while a large $\mu_0$ enforces the toxicity constraint aggressively from the beginning.

3. **Max inner loops** determines the number of optimization steps taken before updating $\lambda$ and $\mu$. This value controls the trade-off, where too few steps may lead to an under-optimized solution, whereas too many leads to wasted compute and runtime.

The main trends are summarized in Figure 7 and exhaustively detailed in Table 3.

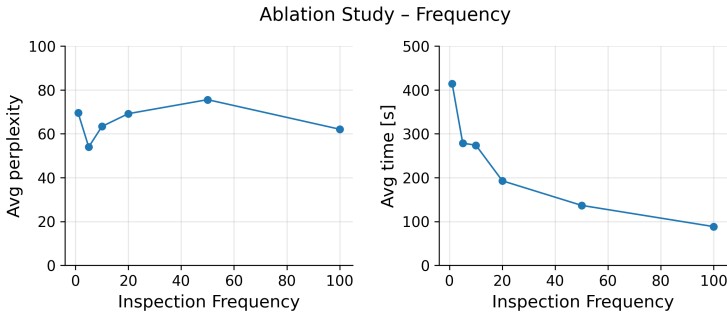

Figure 8: Effect of constraint-checking frequency on runtime and perplexity.

**Frequency Ablation**   We further perform an ablation study with different inspection frequencies. This ablation examines the trade-off between sample quality, constraint satisfaction, and runtime. A more frequent schedule results in consistent adherence at a price of increased compute and runtime. We observe as the inspection frequency increases the average runtime decreases consistently. By analyzing this we demonstrate this method's robustness and note that for all settings, constraint satisfaction is met. Results are shown in Figure 8.

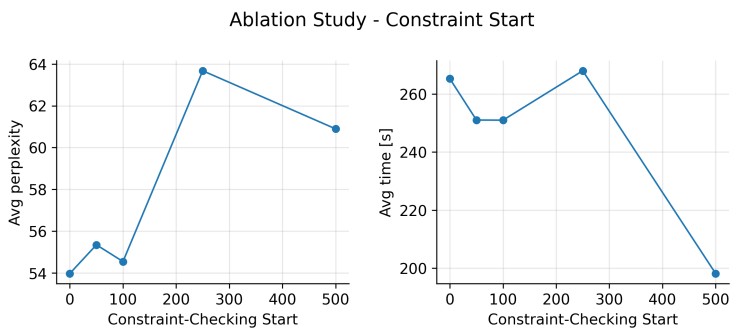

Figure 9: Constraint-checking start analysis on perplexity and runtime.

**Constraint Start Ablation**   We include a constraint-start ablation, where we examine the runtime/perplexity tradeoff by turning the projection operator on only after specific amounts of reverse-diffusion steps. Early enforcements allows the projection to satisfy the constraints at very early iterations at the cost of increased runtime. Notably, across all setting, constraint satisfaction is met for all samples.

# D   Runtime Analysis

To benchmark the computational overhead of CDD as compared to the baselines, we compare the runtime of our method and the other baselines on the toxicity mitigation task. To normalize for the overhead of base models (MDLM and GPT2), we make two adjustment: we reduce the number of diffusion steps from 1000 to 100 and adjust the generation sequence length to 1024; prior work on discrete diffusion has indicated that these hyperparameters can be used to approximately match the inference time of autoregressive models [9]. Empirically, we find that, while the speed is not identical, these adjustments result in a much more similar tokens-per-second rate for the base models. Practically, MDLM could be accelerated even further by reducing the number of diffusion steps further, but we abstain from this so as to maintain consistency with the baselines. For every system we report the mean generation time, the 25th/50th/75th percentiles, and the slowdown relative to the respective baseline.

| ALM hyper-parameters | | | Perplexity | | | | Wall-clock time (s) | | | | LLM-Judge |
|---|---|---|---|---|---|---|---|---|---|---|---|
| LR | $\mu_{\text{init}}$ | Loops | Mean | 25% | 50% | 75% | Mean | 25% | 50% | 75% | Score |
| 0.20 | 1 | 10 | 53.97 | 29.91 | 44.33 | 66.18 | 258.52 | 10.75 | 12.29 | 153.54 | 18.59 |
| 0.20 | 1 | 25 | 64.92 | 28.54 | 44.34 | 66.53 | 448.47 | 13.15 | 15.02 | 374.19 | 19.46 |
| 0.20 | 1 | 50 | 61.95 | 31.44 | 45.58 | 74.66 | 637.53 | 13.18 | 14.82 | 454.52 | 20.29 |
| 0.20 | 5 | 10 | 60.25 | 29.15 | 45.31 | 71.67 | 236.39 | 13.49 | 15.74 | 130.67 | 20.09 |
| 0.20 | 5 | 25 | 59.78 | 28.56 | 44.38 | 72.20 | 343.61 | 10.54 | 11.92 | 192.14 | 19.64 |
| 0.20 | 5 | 50 | 79.71 | 28.32 | 44.53 | 71.44 | 809.47 | 10.60 | 11.93 | 307.58 | 19.31 |
| 0.20 | 10 | 10 | 57.42 | 27.69 | 42.31 | 75.10 | 194.37 | 10.97 | 13.46 | 105.79 | 19.61 |
| 0.20 | 10 | 25 | 58.64 | 27.45 | 43.73 | 67.97 | 636.73 | 10.56 | 12.15 | 126.59 | 19.25 |
| 0.20 | 10 | 50 | 64.90 | 29.97 | 45.67 | 72.46 | 760.28 | 11.03 | 13.86 | 186.85 | 20.49 |
| 0.40 | 1 | 10 | 64.45 | 30.65 | 45.00 | 74.42 | 149.41 | 10.80 | 12.87 | 113.80 | 20.01 |
| 0.40 | 1 | 25 | 62.11 | 33.07 | 46.49 | 76.91 | 223.79 | 10.55 | 12.07 | 200.18 | 20.44 |
| 0.40 | 1 | 50 | 58.95 | 29.10 | 44.23 | 72.99 | 356.07 | 10.50 | 11.52 | 175.66 | 20.15 |
| 0.40 | 5 | 10 | 58.33 | 28.29 | 43.18 | 66.50 | 124.73 | 10.63 | 11.57 | 104.85 | 20.00 |
| 0.40 | 5 | 25 | 68.55 | 29.19 | 45.34 | 69.38 | 213.77 | 10.66 | 11.76 | 204.08 | 20.11 |
| 0.40 | 5 | 50 | 73.02 | 31.72 | 45.83 | 76.92 | 220.34 | 10.65 | 11.74 | 81.31 | 20.18 |
| 0.40 | 10 | 10 | 69.06 | 29.22 | 45.00 | 73.44 | 136.86 | 10.63 | 12.00 | 95.29 | 19.67 |
| 0.40 | 10 | 25 | 61.22 | 30.51 | 45.34 | 72.76 | 193.70 | 10.77 | 12.01 | 123.57 | 20.11 |
| 0.40 | 10 | 50 | 102.30 | 32.47 | 46.62 | 74.88 | 250.82 | 10.70 | 12.44 | 135.07 | 19.77 |
| 0.60 | 1 | 10 | 66.69 | 30.22 | 45.67 | 74.00 | 106.81 | 10.82 | 11.77 | 94.56 | 19.67 |
| 0.60 | 1 | 25 | 87.59 | 32.07 | 45.77 | 69.40 | 196.40 | 10.77 | 12.04 | 170.36 | 19.10 |
| 0.60 | 1 | 50 | 126.00 | 30.38 | 46.98 | 74.85 | 228.66 | 10.64 | 12.02 | 153.14 | 19.23 |
| 0.60 | 5 | 10 | 65.16 | 31.96 | 44.59 | 72.71 | 97.96 | 10.89 | 12.13 | 71.26 | 19.57 |
| 0.60 | 5 | 25 | 64.33 | 29.15 | 45.80 | 77.38 | 144.82 | 10.76 | 11.87 | 144.06 | 19.34 |
| 0.60 | 5 | 50 | 93.99 | 31.56 | 47.41 | 76.91 | 211.75 | 10.91 | 12.14 | 141.92 | 19.65 |
| 0.60 | 10 | 10 | 65.32 | 30.79 | 46.20 | 77.53 | 87.94 | 11.11 | 12.73 | 50.27 | 19.98 |
| 0.60 | 10 | 25 | 68.26 | 30.51 | 47.78 | 75.92 | 158.64 | 10.76 | 11.96 | 70.02 | 19.27 |
| 0.60 | 10 | 50 | 71.86 | 29.99 | 44.51 | 72.47 | 167.33 | 10.85 | 12.17 | 105.36 | 19.55 |
| 0.80 | 1 | 10 | 74.68 | 31.81 | 47.64 | 78.54 | 96.14 | 10.75 | 11.83 | 50.56 | 18.83 |
| 0.80 | 1 | 25 | 77.26 | 30.62 | 45.77 | 76.91 | 158.28 | 10.77 | 11.77 | 115.30 | 20.57 |
| 0.80 | 1 | 50 | 84.82 | 30.51 | 45.77 | 77.38 | 297.94 | 11.90 | 15.24 | 157.88 | 19.91 |
| 0.80 | 5 | 10 | 68.87 | 29.05 | 44.67 | 70.50 | 85.59 | 14.12 | 16.06 | 38.68 | 19.34 |
| 0.80 | 5 | 25 | 60.04 | 26.73 | 44.05 | 68.03 | 129.18 | 14.84 | 17.11 | 64.22 | 19.60 |
| 0.80 | 5 | 50 | 56.15 | 27.77 | 44.24 | 69.18 | 175.82 | 15.61 | 17.76 | 88.31 | 19.41 |
| 0.80 | 10 | 10 | 74.68 | 28.95 | 45.31 | 74.14 | 88.95 | 15.62 | 17.56 | 65.28 | 19.14 |
| 0.80 | 10 | 25 | 78.71 | 29.99 | 45.67 | 69.43 | 146.18 | 14.99 | 16.95 | 105.86 | 19.41 |
| 0.80 | 10 | 50 | 66.70 | 28.29 | 44.24 | 75.66 | 134.23 | 10.65 | 11.74 | 74.80 | 19.50 |

Table 3: Perplexity and wall-clock time statistics, together with LLM-Judge coherence scores, across all ALM hyper-parameter settings.

| Toxicity – Runtime | | | | | | |
|---|---|---|---|---|---|---|
| Model | Size | Runtime (s) | | | | Increase Factor |
| | | Mean | 25% | 50% | 75% | |
| GPT2 | 124M | 10.31 | 10.28 | 10.32 | 10.36 | – |
| GPT2$_{\text{PPLM}}$ | 124M | 878.47 | 857.38 | 869.69 | 887.41 | 85.21 |
| GPT2$_{\text{FUDGE}_{\lambda=2}}$ | 124M | 1387.28 | 1386.47 | 1387.00 | 1387.73 | 134.56 |
| GPT2$_{\text{FUDGE}_{\lambda=9}}$ | 124M | 1481.94 | 1479.71 | 1480.27 | 1484.47 | 143.74 |
| Llama 3.2 | 1B | 16.43 | 16.38 | 16.44 | 16.46 | – |
| MDLM | 110M | 4.59 | 4.40 | 4.49 | 4.74 | – |
| **CDD$_{\tau=0.25}$ (Ours)** | 110M | 401.15 | 9.09 | 9.67 | 11.71 | 87.02 |
| **CDD$_{\tau=0.50}$ (Ours)** | 110M | 230.89 | 9.19 | 9.62 | 10.46 | 50.08 |
| **CDD$_{\tau=0.75}$ (Ours)** | 110M | 63.71 | 9.20 | 9.56 | 10.21 | **13.82** |

Table 4: Runtime statistics for all evaluated models and respective runtime increase from base model. **Bold** and underlined values mark the best and second-best time increase factor, respectively.

Notably, we find that even without optimizing for runtime (besides selecting specific ALM hyperparameters) CDD provides the *least additional overhead* as compared to other controllable generation methods. Further adjustment (e.g., hot-starting the ALM projection, further reduction in diffusion steps) could be made to increase this margin even further.

## E  Entropy Analysis

For the toxicity application, we additionally evaluate CDD and its unconstrained base model using entropy. As described in [10], entropy measures the diversity of the generated sequence. For a generated sequence of length $L$ with $K$ distinct tokens, let $L_k$ denote the count of token $k$, and define the token probabilities $p_k = L_k/L$. The entropy of this distribution is $-\sum_{k=1}^{K} p_k \log p_k$. Lower entropy indicates that probability mass is concentrated on a few tokens, reflecting a lower diversity, whereas a higher entropy indicates a more uniform token usage and greater generative diversity.

As observed in Table 5, across all thresholds, CDD results in a negligible entropy decrease of 0.59–1.0% compared to the base model. As the decrease is small and within narrow confidence intervals, this validates that CDD preserves generative diversity.

| Model | Size | Mean Entropy | 95% CI | % Decrease |
|---|---|---|---|---|
| MDLM | 110M | 5.577 | 0.0178 | – |
| **CDD**$_{\tau = 0.25}$ | 110M | 5.521 | 0.0709 | 1.00 |
| **CDD**$_{\tau = 0.50}$ | 110M | 5.519 | 0.0851 | 1.04 |
| **CDD**$_{\tau = 0.75}$ | 110M | 5.544 | 0.0709 | 0.59 |

Table 5: CDD mean entropy, 95% confidence intervals, and percent decrease from base model at different thresholds for the toxicity application.

## F  Missing Proofs

**Proof of Theorem 4.1**

*Proof.* We begin by proving this bound holds for projected diffusion methods operating in the image space:

$$D_{\mathrm{KL}}(\boldsymbol{x}'_s, \boldsymbol{C}) \;\leq\; (1 - \boldsymbol{\alpha}_t)\, D_{\mathrm{KL}}(\boldsymbol{x}'_t, \boldsymbol{C}) + \boldsymbol{\alpha}_{t+1}^2 G^2, \tag{9}$$

For ease of notation, we will denote subsequent timestep in terms of an arbitrary $t$, such that $\boldsymbol{x}_{t-1} = \boldsymbol{x}_s$ and the subsequent timestep after that is denoted $\boldsymbol{x}_{t-2}$, etc.

Consider that at each iteration of the denoising process, projected diffusion methods can be split into two steps:

1. **Gradient Step:** $\boldsymbol{x}'_t = \boldsymbol{x}_t + \gamma_t \underbrace{\nabla_{\boldsymbol{x}_t} \log q_t(\boldsymbol{x}_t)}_{s_t}$

2. **Projection Step:** $\boldsymbol{x}_{t-1} = \mathcal{P}_{\mathbf{C}}(\boldsymbol{x}'_t)$

These steps are sequentially applied in the reverse process to sample from a constrained subdistribution.

$$\boldsymbol{x}_t \to \overbrace{\boldsymbol{x}_t + \gamma_t s_t}^{\boldsymbol{x}'_t} \to \mathcal{P}_{\mathbf{C}}(\boldsymbol{x}'_t) = \boldsymbol{x}_{t-1} \to \overbrace{\boldsymbol{x}_{t-1} + \gamma_{t-1} s_{t-1}}^{\boldsymbol{x}'_{t-1}} \to \mathcal{P}_{\mathbf{C}}(\boldsymbol{x}'_{t-1}) = \boldsymbol{x}_{t-2} \dots$$

By construction, $\boldsymbol{x}_{t-1} = \mathcal{P}_{\mathbf{C}}(\boldsymbol{x}'_t) \in \mathbf{C}$. Next, let us define the projection distance to $\mathbf{C}$ as:

$$f(\boldsymbol{x}) = D_{\mathrm{KL}}(\boldsymbol{x}, \mathbf{C}) = D_{\mathrm{KL}}\left(\boldsymbol{x} \| \mathcal{P}_{\mathbf{C}}(\boldsymbol{x})\right)$$

Since $\mathbf{C}$ is $\beta$-prox regular, by definition the following hold:

- $f$ is differentiable outside $\mathbf{C}$ (in a neighborhood)

- $\nabla f(\boldsymbol{x}) = 2(\boldsymbol{x} - \mathcal{P}_{\mathbf{C}}(\boldsymbol{x}))$

- $\nabla f$ is $L$-Lipshitz with $L = \frac{2}{\beta}$

The standard "descent lemma" (or smoothness inequality) for $L$-smooth functions applies:

**Lemma F.1.** $\forall \boldsymbol{x}, \boldsymbol{y}$ *in the neighborhood of* $\mathbf{C}$*:*

$$f(\boldsymbol{y}) \leq f(\boldsymbol{x}) + \langle \nabla f(\boldsymbol{x}), \boldsymbol{y} - \boldsymbol{x} \rangle + \frac{L}{2}\|\boldsymbol{y} - \boldsymbol{x}\|^2 = \boxed{f(\boldsymbol{x}) + 2\langle \boldsymbol{x} - \mathcal{P}_{\mathbf{C}}(\boldsymbol{x}), \boldsymbol{y} - \boldsymbol{x}\rangle + \frac{1}{\beta}\|\boldsymbol{y} - \boldsymbol{x}\|^2}$$

Applying this lemma, let us use $\boldsymbol{x} = \boldsymbol{x}'_{t-1}$ and $\boldsymbol{y} = \boldsymbol{x}'_t$. Noting that $\mathcal{P}_{\mathbf{C}}(\boldsymbol{x}'_t) = \boldsymbol{x}_{t-1}$, we get:

$$D_{\mathrm{KL}}(\boldsymbol{x}'_t, \mathbf{C}) \leq \underbrace{D_{\mathrm{KL}}(\boldsymbol{x}'_{t-1}, \mathbf{C})}_{\text{Term A}} + \underbrace{2\,\langle \boldsymbol{x}'_{t-1} - \boldsymbol{x}_{t-2}, \boldsymbol{x}'_t - \boldsymbol{x}'_{t-1}\rangle}_{\text{Term B}} + \underbrace{\frac{1}{\beta}\|\boldsymbol{x}'_t - \boldsymbol{x}'_{t-1}\|^2}_{\text{Term C}} \qquad (\star)$$

**Decomposing Term B.** First, consider that since the step size is decreasing $\gamma_t \geq \gamma_{t-1}$:

$$\boldsymbol{x}'_{t-1} - \boldsymbol{x}_{t-2} \leq (\boldsymbol{x}_{t-1} - \boldsymbol{x}_{t-2}) + \gamma_{t-1}s_{t-1}$$
$$\leq (\boldsymbol{x}_{t-1} - \boldsymbol{x}_{t-2}) + \gamma_t s_{t-1}$$

By the same rationale,

$$\boldsymbol{x}'_t - \boldsymbol{x}'_{t-1} \leq (\boldsymbol{x}_t - \boldsymbol{x}_{t-1}) + \gamma_t(s_t - s_{t-1}). \qquad \text{(Definition B.1)}$$

*Proof of non-expansiveness of the projection operator.* Next, we prove the non-expansiveness of the projection operator:

$$\|\boldsymbol{x}_t - \boldsymbol{x}_{t-1}\| \leq 2\,\gamma_{t+1}G^2 \qquad (\mathcal{L}^+)$$

Given $\boldsymbol{x}_t = \mathcal{P}_{\mathbf{C}}(\boldsymbol{x}'_{t+1})$ and $\boldsymbol{x}_{t-1} = \mathcal{P}_{\mathbf{C}}(\boldsymbol{x}'_t)$,

$$\|\boldsymbol{x}_t - \boldsymbol{x}_{t-1}\| = \|\mathcal{P}_{\mathbf{C}}(\boldsymbol{x}'_{t+1}) - \mathcal{P}_{\mathbf{C}}(\boldsymbol{x}'_t)\| \leq \|\boldsymbol{x}_{t+1} - \boldsymbol{x}_t\|$$

since projections onto closed prox-regular sets are $L$-Lipshitz.

Now:

$$\boldsymbol{x}'_{t+1} = \boldsymbol{x}_{t+1} + \gamma_{t+1}s_{t+1};$$
$$\boldsymbol{x}'_t = \boldsymbol{x}_t + \gamma_t s_t;$$
$$\boldsymbol{x}'_{t+1} - \boldsymbol{x}'_t = (\boldsymbol{x}_{t+1} - \boldsymbol{x}_t) + (\gamma_{t+1}s_{t+1} - \gamma_t s_t). \qquad \text{(Definition B.2)}$$

Making the projection residual,

$$\boldsymbol{x}_{t+1} - \boldsymbol{x}_t = \boldsymbol{x}_{t+1} - \mathcal{P}_{\mathbf{C}}(\boldsymbol{x}'_{t+1})$$

orthogonal to the target space at $\boldsymbol{x}_t$ (and any vector of the form $s_{t+1} - s_t$). Thus, since $\|s_t\| \leq G \ \ \forall t$:

$$\|\boldsymbol{x}'_{t+1} - \boldsymbol{x}'_t\|^2 = \|\gamma_{t+1}s_{t+1}\|^2 + \|\gamma_t s_t\| \leq (\gamma_{t+1}^2 + \gamma_t^2)G^2$$

Taking the square root:

$$\|\boldsymbol{x}'_{t+1} - \boldsymbol{x}'_t\| \leq \sqrt{\gamma_{t+1}^2 + \gamma_t^2}G$$

Since $\gamma_{t+1} \geq \gamma_t$:

$$\|\boldsymbol{x}'_{t+1} - \boldsymbol{x}'_t\| \leq \sqrt{2}\gamma_{t+1}G$$
$$< 2\gamma_{t+1}G$$

Finally, by applying Definition (B.1), $\|\boldsymbol{x}_{t+1} - \boldsymbol{x}_t\| \leq \|\boldsymbol{x}'_{t+1} - \boldsymbol{x}'_t\|$, and thus:

$$\boxed{\|\boldsymbol{x}_{t+1} - \boldsymbol{x}_t\| \leq 2\gamma_{t+1}G}$$

$\square$

Now, prox-regularity gives:
$$\langle \boldsymbol{x}'_{t-1} - \boldsymbol{x}_{t-2}, \boldsymbol{x}'_t - \boldsymbol{x}'_{t-1} \rangle \le \beta \|\boldsymbol{x}_t - \boldsymbol{x}_{t-1}\|^2$$
$$\le 4\beta \gamma_{t+1}^2 G^2 \qquad \text{(Bound B.1)}$$

where the Bound B.1 is derived by applying $(\mathcal{L}^+)$.

Since $\mathbf{C}$ in $\beta$-prox regular, for any point $u$ near $\mathbf{C}$ and $v \in \mathbf{C}$:
$$\langle u - \mathcal{P}_{\mathbf{C}}(u), v - \mathcal{P}_{\mathbf{C}}(u) \rangle \le \beta \|v - \mathcal{P}_{\mathbf{C}}(u)\|^2$$

Above, we substitute:
$$u = \boldsymbol{x}'_t = \boldsymbol{x}_t + \gamma_t s_t$$
$$v = \boldsymbol{x}_t$$
$$\mathcal{P}_{\mathbf{C}}(u) = \boldsymbol{x}_{t-1}$$

Now, expanding the inner product:

$$
\begin{aligned}
\langle \boldsymbol{x}'_{t-1} - \boldsymbol{x}_{t-2}, \boldsymbol{x}'_t - \boldsymbol{x}'_{t-1} \rangle &= \langle \boldsymbol{x}'_{t-1} - \boldsymbol{x}_{t-2}, (\boldsymbol{x}_t + \gamma_t s_t) - (\boldsymbol{x}_{t-1} + \gamma_{t-1} s_{t-1}) \rangle \\
&\le \langle \boldsymbol{x}'_{t-1} - \boldsymbol{x}_{t-2}, (\boldsymbol{x}_t - \boldsymbol{x}_{t-1}) + \gamma_t (s_t - s_{t-1}) \rangle \\
&\le \langle \boldsymbol{x}'_{t-1} - \boldsymbol{x}_{t-2}, (\boldsymbol{x}_t - \boldsymbol{x}_{t-1}) \rangle + \langle \boldsymbol{x}'_{t-1} - \boldsymbol{x}_{t-2}, \gamma_t (s_t - s_{t-1}) \rangle
\end{aligned}
$$

and since $\|s_t\| \le G \quad \forall t : \langle s_{t+1}, s_t \rangle \le \|s_{t+1}\| \|s_t\| \le G^2$ so $\langle s_{t-1}, s_t \rangle - \|s_{t+1}\|^2 \le G^2$, and:
$$\langle \boldsymbol{x}'_{t-1} - \boldsymbol{x}_{t-2}, \gamma_t (s_t - s_{t-1}) \rangle \le \gamma_t^2 G^2 \qquad \text{(Bound B.2)}$$

By applying Definition (B.2):
$$
\begin{aligned}
\langle \boldsymbol{x}'_{t-1} - \boldsymbol{x}_{t-2}, \boldsymbol{x}'_t - \boldsymbol{x}'_{t-1} \rangle &= \langle \boldsymbol{x}'_{t-1} - \boldsymbol{x}_{t-2}, (\boldsymbol{x}_t - \boldsymbol{x}_t) + (\gamma_t s_t - \gamma_{t-1} s_{t-1}) \rangle \\
&\le \langle \boldsymbol{x}'_{t-1} - \boldsymbol{x}_{t-2}, (\boldsymbol{x}_t - \boldsymbol{x}_{t-1}) \rangle
\end{aligned}
$$

Therefore, by applying Bound (B.1) to the previous inequality and Bound (B.2) directly, Term B is upper bounded by:

$$\boxed{2 \langle \boldsymbol{x}'_{t-1} - \boldsymbol{x}_{t-2}, \boldsymbol{x}'_t - \boldsymbol{x}'_{t-1} \rangle \le 8\beta \gamma_{t+1}^2 G^2 + 2\gamma_t^2 G^2} \qquad \text{(Bound B.3)}$$

**Decomposing Term C.** Next, we derive a bound on Term C in Eq. $(\star)$. As already shown,
$$\|\boldsymbol{x}'_t - \boldsymbol{x}'_{t-1}\| \le 4\gamma_t G,$$

given:
$$\boldsymbol{x}'_t - \boldsymbol{x}'_{t-1} \le \underbrace{(\boldsymbol{x}_t - \boldsymbol{x}_{t-1})}_{\le 2\gamma_{t+1} G} + \underbrace{\gamma_t (s_t - s_{t-1})}_{\le 2G}$$
$$\le 4\gamma_{t+1} G$$

Thus,

$$\boxed{\frac{1}{\beta} \|\boldsymbol{x}'_t - \boldsymbol{x}'_{t-1}\|^2 \le \frac{16}{\beta} \gamma_{t+1}^2 G^2} \qquad \text{(Bound C.1)}$$

Combining bounds (B.3) and (C.1) into $(\star)$, and recalling that $\gamma_{t+1} \ge \gamma_t$:
$$D_{\mathrm{KL}}(\boldsymbol{x}'_t, \mathbf{C}) \le \underbrace{D_{\mathrm{KL}}(\boldsymbol{x}'_{t-1}, \mathbf{C})}_{d} + \underbrace{(8\beta + 2 + \frac{16}{\beta}) \gamma_{t+1}^2 G^2}_{K}$$

Now, we rewrite Term A, which for ease of notation we will refer to as $d$:
$$d = (1 - 2\beta \gamma_{t+1})d + 2\beta \gamma_{t+1} d$$

Thus:
$$
\begin{aligned}
D_{\mathrm{KL}}(\boldsymbol{x}'_t, \mathbf{C}) &\le d - 2\beta \gamma_{t+1} d + 2\beta \gamma_{t+1} d + K \gamma_{t+1}^2 G^2 \\
&= (1 - 2\beta \gamma_{t+1})d + \left[ 2\beta \gamma_{t+1} d + K \gamma_{t+1}^2 G^2 \right]
\end{aligned}
$$

Next, through Young's inequality, we simplify this expression further.

**Theorem F.2. (Young's Inequality)** $\forall u, v \geq 0, \epsilon > 0$:

$$uv \leq \frac{u^2}{2\epsilon} + \frac{\epsilon v^2}{2}$$

If we choose $u = \sqrt{2\beta\gamma_{t+1}}d$, $v = \sqrt{K}\gamma_{t+1}G$, and $\epsilon = \frac{2\beta d}{k\gamma_{t+1}G^2}$, then

$$\begin{aligned} uv &= \sqrt{2\beta\gamma_{t+1}}d \times \sqrt{K}\gamma_{t+1}G \\ &= \sqrt{2K}\gamma_{t+1}^{\frac{3}{4}}Gd \end{aligned}$$

Applying Young's Inequality:

$$\begin{aligned} uv &\leq \frac{u^2}{2\epsilon} + \frac{\epsilon v^2}{2} \\ &= \frac{2\beta\gamma_{t+1}d}{2(\frac{2\beta d}{K\gamma_{t+1}G^2})} + \frac{\epsilon v^2}{2} \\ &= \frac{K\gamma_{t+1}^2 G^2}{2} + \frac{\epsilon v^2}{2} \\ &= \frac{K\gamma_{t+1}^2 G^2}{2} + \left( \frac{1}{2} \times \frac{2\beta d}{K\gamma_{t+1}G^2} \times K\gamma_{t+1}^2 G^2 \right) \\ &= \frac{K\gamma_{t+1}^2 G^2}{2} + \beta\gamma_{t+1}d \end{aligned}$$

Thus,

$$\sqrt{2K}\gamma_{t+1}^{\frac{3}{4}}Gd \leq \frac{K\gamma_{t+1}^2 G^2}{2} + \beta\gamma_{t+1}d$$

Finally, taken altogether:

$$\begin{aligned} 2\beta\gamma_{t+1}d + K\gamma_{t+1}^2 G^2 &\leq \beta\gamma_{t+1}d + \left( \frac{K\gamma_{t+1}^2 G^2}{2} + \beta\gamma_{t+1}d \right) \\ &= 2\beta\gamma_{t+1}d + \frac{K}{2}\gamma_{t+1}^2 G^2 \end{aligned}$$

Since $\gamma_{t+1} \leq \frac{\beta}{2G^2}$, then

$$\frac{K}{2}\gamma_{t+1}^2 G^2 \leq \frac{1}{2}\left( 8\beta + 2 + \frac{16}{\beta} \right)\frac{\beta^2}{4G^2} = \mathcal{O}(\beta^3)$$

which is bounded by $\gamma_{t+1}^2 G^2$ for all $\beta \geq 0$.

Thus,

$$2\beta\gamma_{t+1}d + K\gamma_{t+1}^2 G^2 \leq 2\beta\gamma_{t+1}d + \gamma_{t+1}^2 G^2.$$

By substitution we obtain:

$$D_{\mathrm{KL}}(\boldsymbol{x}_t', \mathbf{C}) \leq (1 - 2\beta\gamma_{t+1})D_{\mathrm{KL}}(\boldsymbol{x}_{t-1}', \mathbf{C}) + \underbrace{\gamma_{t+1}^2 G^2}_{\mathcal{O}(\beta^3)}$$

Finally, we reparameterize this, such that $\boldsymbol{\alpha}_t = 2\beta\gamma_t$, implying $\gamma_t = \frac{\alpha_t}{2\beta}$. Plugging this in, we get:

$$\begin{aligned} D_{\mathrm{KL}}(\boldsymbol{x}_t', \mathbf{C}) &\leq (1 - \boldsymbol{\alpha}_t)D_{\mathrm{KL}}(\boldsymbol{x}_{t-1}', \mathbf{C}) + \gamma_{t+1}^2 G^2 \\ &\leq (1 - \boldsymbol{\alpha}_t)D_{\mathrm{KL}}(\boldsymbol{x}_{t-1}', \mathbf{C}) + \boldsymbol{\alpha}_{t+1}^2 G^2 \end{aligned}$$

and thus,

$$\boxed{D_{\mathrm{KL}}(\boldsymbol{x}_t', \mathbf{C}) \leq (1 - \boldsymbol{\alpha}_t)D_{\mathrm{KL}}(\boldsymbol{x}_{t-1}', \mathbf{C}) + \boldsymbol{\alpha}_{t+1}^2 G^2}$$

$\square$

