# OpenReview forum: "Constrained Discrete Diffusion"
_NeurIPS.cc/2025/Conference — NeurIPS 2025 poster_

### Official Review · Reviewer_c5DD · 2025-06-27

**Clarity:** 3
**Significance:** 4
**Originality:** 4
**Rating:** 5
**Confidence:** 3

**Summary:**

This work presents an inference time algorithm for controlled generation with discrete diffusion models to produce samples that satisfy user-specified constraints. Importantly, the proposed constraint discrete diffusion (CDD) algorithm can be applied on top of pre-trained models without requiring fine-tuning or re-training of the underlying denoising model.

**Questions:**

1. For the counting constraint task, having access to the ground truth correct count response seems to render the results here less interpretable. Could the authors clarify why this is not label leakage for this task?

**Ethical Concerns:**

["NO or VERY MINOR ethics concerns only"]

**Final Justification:**

This is solid work that would be of interest to the community. During the rebuttal, the authors addressed my concerns.

**Limitations:**

The increased computational complexity at inference time is described in Section E.

**Quality:**

3

**Strengths And Weaknesses:**

### **Strengths**

The proposed method is a novel adaptation of a comparable method for continuous signal diffusion models from Christopher et al [1]. The ability to layer this CDD algorithm on top of pre-trained models without the need to perform additional training renders the approach especially appealing in my opinion and aligns with recent emphasis in our field on leveraging test time compute / inference scaling.

The experimental section is robust and the results are convincing. The details provided in the appendix are extensive and indicative of the reproducibility of this method.

### **Weaknesses**

1. The notation used in Sections 3 and 4 was confusing in my opinion. Specifically, in Section 3 the boldface variables, e.g., $\mathbf{x}\_s, \mathbf{x}\_t$ represent one-hot variables. However, if my understanding is correct then in Section 4, these variables, e.g., $\mathbf{x}\_{t(\ell)}, \mathbf{x}\_s, \mathbf{y}$, seem to represent vectors in the interior of the simplex, i.e. not one-hot vectors.
2. It would be important to add entropy to tables in Figure 3 and Table 1 to ensure the generative perplexity value is not being “gamed”, an issue several works have noted with this metric, e.g., Zheng et al [2].
3. Along those lines, the drop in quality (as measured by PPL) in the table in Figure 3 between MDLM and CDD is quite stark.
4. In Appendix A, the authors note that there are no open-source large scale discrete diffusion models, however there have been several released recently, specifically Llada [3] and Dream [4] which are both 7-8B parameter models.


### **Other / More Minor Comments & Suggestions**

1. I found the intro to be quite heavily focused on the molecule generation application of this method, but the experimental results clearly indicate that this is a more widely applicable algorithm.
2. On Line 152, the citation format for “Xu et al” seems to be inconsistent with the “[xx]” used throughout the rest of the manuscript. (Same issue on line 843)
3. Line 297 is missing the word “Appendix” before “Appendix C”.
4. There is a typo on line 335, “scientifi tasks” should be “scientific tasks”.

---

[1] Christopher, Jacob K., Stephen Baek, and Nando Fioretto. "Constrained synthesis with projected diffusion models." Advances in Neural Information Processing Systems 37 (2024): 89307-89333.

[2] Zheng, Kaiwen, et al. "Masked diffusion models are secretly time-agnostic masked models and exploit inaccurate categorical sampling." arXiv preprint arXiv:2409.02908 (2024).

[3] Nie, Shen, et al. "Large language diffusion models." arXiv preprint arXiv:2502.09992 (2025).

[4] Jiacheng, Ye et al. “Dream 7B”

---

> ### Author Rebuttal · Authors · 2025-07-31
>
> We thank the reviewer for their insights, particularly their acknowledgment of our robust experimental results as well as the timeliness of our work. Below, we provide our answers to their questions and comments.
>
> > **1. The notation used in Sections 3 and 4 was confusing in my opinion.**
>
> We are happy to clarify. The variables represent vectors in the interior of the simplex at higher noise levels. As $t \rightarrow 0$, these vectors transition to one-hots. For instance, when MDLM unmasks a token, it transitions to a one-hot; the masking probability decreases with $t$, so the probability of a given position being a one hot changes as dictated by the masking schedule.
>
> We can expand further on the notation in the final version of the paper and we are certainly also receptive to the reviewer's suggestions they may provide during rebuttal.
>
>
> > **2. It would be important to add entropy to tables in Figure 3 and Table 1 to ensure the generative perplexity value is not being “gamed”, an issue several works have noted with this metric, e.g., Zheng et al [2].**
>
> Excellent point. To check whether the true generative perplexity may be "gamed" we have ran some additional experiments during rebuttal and include an LLM-as-a-Judge as an evaluation metrics. _Interestingly, it shows no degradation in generation quality_.
>
> | **Setting**   | **Avg Entropy** | **95% CI** |
> |---------------|------------------|------------|
> | MDLM       | 5.577            | 0.017823   |
> | $\text{CDD}_{\tau=0.75}$ | 5.544            | 0.070966   |
> |$\text{CDD}_{\tau=0.50}$ | 5.519            | 0.085119   |
> | $\text{CDD}_{\tau=0.25}$ | 5.521            | 0.085564   |
> |$\text{GPT2}_\text{PPLM}$  | 2.978            | 0.166751   |
>
> In the table above, we have show results for further experimentation with the entropy-based evaluation as described in [A] and report that CDD preserves predictive entropy relative to its base diffusion model (MDLM in this case) with at most a ∼ 1% decrease in average entropy. This demonstrates that CDD preserves diversity even while enforcing toxicity constraints. The last row of the table reports yet another model we ran during rebuttal, the baseline of PPLM for Autoregressive models, which sees a much larger decrease in token diversity!
>
> We are happy to include and extend these results in the final version of the paper.
>
>
> > **3. In Appendix A, the authors note that there are no open-source large scale discrete diffusion models, however there have been several released recently, specifically Llada [3] and Dream [4] which are both 7-8B parameter models**
>
> We are quite excited to see the significant progress that researchers have recently made in the development of discrete diffusion for language generation! Indeed, we will modify this statement to indicate that **no open-source large scale discrete diffusion models were available at the time of development**. In fact, Dream [A] was released in April, and Llada [B] appeared in late February, with the code being released several weeks later.
>
> We are excited to extend CDD to these larger architectures in our future work, and we are currently actively working on it.
>
>
> > **3. Along those lines, the drop in quality (as measured by PPL) in the table in Figure 3 between MDLM and CDD is quite stark.**
>
> While we acknowledge that CDD does incur a perplexity decrease relative to the base MDLM, there are many scenarios where this as an acceptable trade-off for strict constraint satisfaction, especially in safety-critical settings where any violation is unacceptable. Furthermore, controllable generation baselines like PPLM and FUDGE demonstrate an even more substantial perplexity degradation while still resulting in constraint violations.
>
> As shown in Response (2), while other controllable generation methods may be "gaming" the perplexity metric, CDD certainly is not. *Thus, (and as you also acknowledged) the perplexity metric is not a perfect gauge of the model's true performance*. As result we further use LLM-as-a-Judge as a quality metric in which CDD actually benefits with a minimal improvement, while PPLM and FUDGE report large degradation. Additionally, when measuring entropy, another important quality metric, we report negligible decrease in token diversity.
>
>
>
> > **4. On Line 152, the citation format for “Xu et al” seems to be inconsistent with the “[xx]” used throughout the rest of the manuscript. (Same issue on line 843) | Line 297 is missing the word “Appendix” before “Appendix C”. | There is a typo on line 335, “scientifi tasks” should be “scientific tasks”.**
>
> Thank you very much! We have updated our paper with these fixes.
>
>
>
> ---
>
> We hope the clarifications above resolve the remaining questions about our contributions, and we are happy to elaborate further as needed. Thank you!
>
>
> [A] Jiacheng, Ye et al. “Dream 7B”
>
> [B] Nie, Shen, et al. "Large language diffusion models." arXiv preprint arXiv:2502.09992 (2025).
>
> [C] Sahoo, Subham, et al. "Simple and effective masked diffusion language models." Advances in Neural Information Processing Systems 37 (2024): 130136-130184.
>
> [D] Lou, Aaron, Chenlin Meng, and Stefano Ermon. "Discrete diffusion modeling by estimating the ratios of the data distribution." arXiv preprint arXiv:2310.16834 (2023).

---

> > ### Comment · Reviewer_c5DD · 2025-07-31
> >
> > Thank you for the detailed response. I maintain my score as I believe this is a nice contribution that builds off the recent success in discrete diffusion models.
> >
> > Most of my questions have been answered except:
> > - I still maintain that the notation between Section 3 and 4 is hard to parse and deviates from what other papers, e.g., MDLM and UDLM, use to denote latent vectors. I believe the authors are conflating notation for realizations of the latent variables (e.g. one hot / noised tokens) vs. distributions over these latent variables. Perhaps to other readers this distinction can be clear from the context but to me it was not.
> > - The authors seem to have missed my question about the counting experiment.

---

> > > ### Author Response · Authors · 2025-08-01
> > >
> > > Thank you for your continued support of our work and your willingness to engage in the discussion period. Below we respond to the previous point we missed during our initial rebuttal.
> > >
> > > > **I still maintain that the notation between Section 3 and 4 is hard to parse...**
> > >
> > > This point is well taken, and we will certainly take this suggestion seriously. We are more than happy to add more explicit clarifications in an updated version of the manuscript. Thank you for your attention to this detail!
> > >
> > >
> > > > **For the counting constraint task, having access to the ground truth correct count response seems to render the results here less interpretable. Could the authors clarify why this is not label leakage for this task?**
> > >
> > > Thanks you for raising this point as we seem to have overlooked it in our original response. Label leakage typically occurs when ground-truth labels are inadvertently revealed to the model during training or inference. Let us begin by emphasizing that there is no overlap between our validation/test sets and the training set with which was used for instruction-tuning the models. Hence, the validation/test set labels are never seen by the model during training.
> > >
> > > Although our custom projection module accurately recovers the ground-truth counts, it does this without access to the labels. Specifically, our projection operator is comprised of a custom rule-based constraint that employs an external counting module; however, this module is independent and has no direct knowledge of the ground-truth labels. While the result computed by this module effectively matches the ground-truth (as reflected by perfect constraint satisfaction), this is accomplished by an independent reasoning component, which in no way accesses the true labels.
> > >
> > > Ultimately, this experiment exemplifies the generality of the method to accommodate custom rule-based projection algorithms.
> > >
> > > ---
> > >
> > > We hope this answers your question but as always we are happy to engage in further discussion!

---

> ### Comment · Reviewer_c5DD · 2025-08-04
>
> Thank you for the additional clarifications.
>
> On this point:
> > _Although our custom projection module accurately recovers the ground-truth counts, it does this without access to the labels_
>
> So the counting module is an external (differentiable?) classifier that takes in the sequence and returns the count of the letters?

---

> > ### Author Response · Authors · 2025-08-04
> >
> > Thanks for the question. We're happy to clarify!
> >
> > > **So the counting module is an external (differentiable?) classifier that takes in the sequence and returns the count of the letters?**
> >
> > Not exactly. To be very specific, our counting module is a simple rule-based operator that considers the word and character in question and applies a binary mask across a unicode representation of the word. The mask indicates which characters match the character in question. We then apply a sum operation across this mask to compute the actual value. We then tokenize this value, and our differentiable function $\Delta g$ simply enforces that the $\arg \max$ of the probability distribution matches that particular token, essentially matching the penalty function used lexical constraint. We provide more details about this in Appendix C.3.1, and, to improve clarity, will add description of this connection to the lexical constraint in our next revision.
> >
> > Thank you for your interest!

---

> > > ### Comment · Reviewer_c5DD · 2025-08-08
> > >
> > > Thank you for responding. Perhaps I am still misunderstanding something but to me it sounds like your "external module" is just providing the ground truth label and is therefore label leakage. That is, if we gave any model the task
> > > > "How many "r"s are in the word "strawberry"? Your answer should say 3.
> > >
> > > then the model would output the correct answer (I am assuming).
> > >
> > > In any case, I don't think this confusion is "blocking" in any sense and the paper remains a strong work with important contributions.

---

> > > > ### Author Response · Authors · 2025-08-09
> > > >
> > > > Thank you, again, for your appreciation of our work and the engaging discussion!
> > > >
> > > > Regarding the counting settings, perhaps this would be better explained with an illustrative example: Consider the example response in Figure 1. Before projecting, the $\arg\max$ of the predicted sequence may look exactly like the baseline:
> > > >
> > > > > There are 2 R's in "strawberry"
> > > >
> > > > Examining the probability distribution of the numerical token, the top-k may look something like this:
> > > >
> > > > > ["2": 0.4, "3": 0.3, "4": 0.1, ...]
> > > >
> > > > Our external module, which can be viewed as a *neuro-symbolic oracle* for this task, would then be queried and determine that the correct token is actually "3".
> > > > \
> > > > The probability distribution would then be adjusted:
> > > >
> > > > > ["3": 0.41, "2": 0.33 "4": 0.10, ...]
> > > >
> > > > After renormalizing the probability distribution (e.g., enforcing that all probabilities add to 1), the projected distribution is realized.
> > > >
> > > > Now, let us stress an important distinction; while the external module does indeed compute the correct token, it does so agnostically to any ground-truth labels. The external module, in this specific case, arrives at the correct value through a rule-based mechanism, rather than via access to the true label directly.
> > > > \
> > > > Irregardless, the motivation behind this experiment is to demonstrate that the projection can be facilitated in many different ways -- not just through surrogate-driven assessment of the constraints but also through hard, rule-based operators.
> > > >
> > > > We sincerely appreciate this engagement. Again, many thanks for your valuable input and for championing our work!

---

### Official Review · Reviewer_3U4L · 2025-06-27

**Clarity:** 1
**Significance:** 2
**Originality:** 2
**Rating:** 4
**Confidence:** 3

**Summary:**

The paper introduces a method for enforcing constraints during inference in discrete diffusion models. It builds on the framework of concrete score matching, used here as a discrete denoising strategy. The core idea is to incorporate a constraint set CC into the denoising process to guide generation. However, two main challenges arise: 1) The constraints can only be meaningfully evaluated on fully denoised instances, 2)The constraint set C is intractable; one can only evaluate whether a candidate $x \in C$.

To address these issues, the authors propose projecting the intermediate noisy instance to the fullyb denoised distribution and  relaxing the constraints into an optimization objective. This is implemented using an augmented Lagrangian projection, which allows for approximate enforcement of constraints during inference.

Empirical results across two domains, language modeling and molecular generation, demonstrate the effectiveness of the proposed approach in improving constraints enforcement.

**Questions:**

## Suggestions for Improvement

I think the following suggestions would greatly contribute to improve the paper:
- Include a concise but self-contained explanation of concrete score matching or, at minimum, provide intuition and references.
- Discuss the choice and role of the noise distribution $\nu$ used in training and inference.
- Reorganize the structure so that preliminary material does not dominate sections meant for methodological exposition.
- If relevant, discuss the relation of the method to RL-based constraint enforcement approaches, and provide comparative context.
- Justify the use of concrete score matching over $x_0$-prediction in terms of both modeling power and practical advantages.

## Question:
Why not use a more common $x_0$-prediction-based method?

**Ethical Concerns:**

["NO or VERY MINOR ethics concerns only"]

**Final Justification:**

hTe authors have addressed most of my concerns during the discussion. They have committed to substantial revisions in the updated version of the paper. My score reflects the fact  these forthcoming changes cannot be evaluated.

**Limitations:**

The proposed method includes an optimization step at each denoising step, which suggests a potentially significant computational overhead. This limitation is not discussed in the paper. A thorough discussion, ideally accompanied by empirical evaluation, of the method’s computational cost would greatly improve the assessment of its practical applicability and scalability.

**Quality:**

1

**Strengths And Weaknesses:**

## Strengths

- The use of an augmented Lagrangian formulation to enforce constraints during discrete diffusion is an interesting and potentially impactful idea.
- The method is accompanied by a theoretical justification, which lends formal support to the approach, although, as discussed below, its assumption are difficult to verify.

## Weaknesses

### Lack of Clarity
Several important aspects of the paper remain unclear due to missing definitions, unspecified choices, or inconsistencies across sections. Below are representative examples of clarity issues:

Undefined terms: Equation (5) appears central to understanding the proposed method. However, several critical terms in the equation such as $x'$, $y$, and $t(l)$ are not defined.

External dependencies without context:
- The paper relies on concrete score matching as the core denoising framework, but it is not explicitly described and justified. In contrast, the diffusion-based method using $x_0$-prediction is explicitly explained.
- The $\beta$-prox-regularity assumption in Theorem 4.1 is used without definition or intuition, only a reference. Readers unfamiliar with this property are left without the tools to evaluate its implications.

Unspecified experimental choices: The noise distribution $\nu$ used in experiments is never mentioned. How the choice of noise affect the model's behavior (and which noise choice are even possible within the method is not discussed).

Structural inconsistencies: The first three paragraphs of Section 4 continue to introduce background material rather than presenting core contributions. This disrupts the logical flow of the paper and may confuse readers expecting new material at that stage.

### 2. Missing Discussion of Connections to RL

The proposed method, after constraint relaxation, resembles a reinforcement learning (RL) framework where the augmented Lagrangian term serves as a reward signal. If this interpretation is correct, it raises the following concerns:
- The paper does not position the work in the context of related RL-based constraint enforcement methods.
- There is no comparative discussion or evaluation against RL baselines in either the related work section or experiments.

Given this potential overlap, the absence of any mention of RL frameworks appears to be a missed opportunity for clarity and positioning.

### 3. Unjustified Methodological Choices

- It remains unclear why concrete score matching was chosen as the modeling framework. Since the method requires sampling from $x_0$, it is reasonable to ask: *Why not use a more common $x_0$-prediction-based method*?


### 4. Evaluation Concerns

- It is surprising that the proposed method is not applied to maximize validity, a common metric in molecular generation tasks. This raises concerns that the method may not be effective for enforcing structural validity. At least, the authors should discuss this point.

---

> ### Author Rebuttal · Authors · 2025-07-31
>
> Thank you for the comments. We respond to your questions below and welcome further discussion.
>
> > **1. Undefined terms: Equation (5) appears central to understanding the proposed method. However, several critical terms in the equation such as $\mathbf{x}'$, $\textbf{y}$, and $t(\ell)$ are not defined.**
>
> This is something that might have been overlooked. All the symbols are defined in the paper, e.g., $x_{t(\ell)}'$ is defined in Equation (3), and $\mathbf{y}$ is defined in line 177, it's the minimizer variable in the $\arg\min$ (the projected sequence).
>
>
> > **2. The paper relies on concrete score matching as the core denoising framework, but it is not explicitly described and justified. In contrast, the diffusion-based method using $\mathbf{x_0}$ prediction is explicitly explained. Why not use a more common $\mathbf{x_0}$-prediction-based method?**
>
> Please notice that we describe and justify the use of score matching in the first paragraph of Section 4.
>
> To elaborate further, projected diffusion sampling processes have recently emerged for constrained generation within continuous domains. These approaches have adopted score-based Langevin samplers due to the theoretical perspective that is enabled by this sampling approach. Our work extends this to discrete domains, offering both SOTA empirical results *and* novel strong theoretical guarantees. *Concrete score matching, in particular, enables presenting the theoretical results in a clear way*.
>
> However, we'd like to stress that **$\mathbf{x_0}$ predictive models implicitly capture concrete score matching!** We'd encourage the reviewer to check Appendix C of [D], where the connection between MDLM (the model we employ) and concrete score matching is explicitly drawn.
>
> Also note that **our experimental validation adopts $\mathbf{x_0}$-prediction-based generation**. CDD is parameterization-agnostic, and the projection step only requires the per-step categorical token distributions, which are available under both $\mathbf{x_0}$-prediction and Concrete Score Matching parameterizations. MDLM uses $\mathbf{x_0}$-prediction on masked positions while previously unmasked positions remain fixed, resulting in the distributions that CDD projects onto with a KL-based augmented-Lagrangian optimization.
>
>
> > **3. The $\beta$ -prox-regularity assumption in Theorem 4.1 is used without definition or intuition, only a reference. Readers unfamiliar with this property are left without the tools to evaluate its implications.**
>
> The definition of $\beta$-prox regularity is from [C] (Def. 13.27) as specified already in the statement of Theorem 4.1. This is a common notion in convex optimization theory, but we will be happy to add additional details in the final version of the paper.
> In short, this condition is a relaxation of typical convexity assumptions; it implies that for each viable point and normal direction, small perturbations still project uniquely and smoothly back to $\mathbf{C}$. This is used instead of a convexity assumption, as it **allows us to broaden the applicability of the theory** to constraint sets that are not strictly convex.
>
>
> > **4. Unspecified experimental choices: The noise distribution $\nu$ used in experiments is never mentioned. How the choice of noise affect the model's behavior (and which noise choice are even possible within the method is not discussed)**
>
> This might have been overlooked by the reviewer. Please note that the reference distribution $\nu$ is the fixed categorical corruption used in the forward process as discussed in lines~115--121. As noted we discuss two standard choices:
> (i) mask corruption, where $\nu$ is a one-hot on [MASK] as done in MDLM [A] (better for larger vocabularies such as natural language); and
> (ii) uniform corruption, where $\nu$ is uniform over the vocabulary as in UDLM [B] (better for smaller vocabularies such as SMILES).
>
> In our experiments, natural-language tasks use the corruption choice introduced in MDLM, in which $\nu$ is the one-hot [MASK], while molecule generation uses the corruption introduced in UDLM where $\nu$ is uniform. CDD’s augmented-Lagrangian projection operates on the token-probability vectors returned by the reverse sampler after each denoising step and is therefore compatible with any fixed $\nu$ the base diffusion model is trained with.
>
>
> > **5. Missing Discussion of Connections to RL. The paper does not position the work in the context of related RL-based constraint enforcement methods. There is no comparative discussion or evaluation against RL baselines in either the related work section or experiments.**
>
>
> RL-methods differ fundamentally from our approach in both goal and methodology:
>
> 1. Our method enforces hard, per-sample constraints at inference time in discrete diffusion, without requiring retraining. In contrast, RL methods operate during training and optimize expected reward across samples and do not offer per-sample control at test time.
>
> 2. Our approach projects each diffusion step back onto the feasible set, providing theoretical guarantees and ensuring zero constraint violations in our experiments. RL-based methods rely on policy optimization and do not provide such guarantees during sampling.
>
> CDD intervenes in the sampling process, and **the augmented Lagrangian method enforces constraints by projecting the sample onto the constraint set**, rather than implicitly through reward maximization.
> Recall also that CDD achieves zero violations in our experiments without retraining.
>
> *Because of these fundamental mechanisms and objective mismatch, RL-based approaches are deemed outside the scope of this work.*
> We will be happy to add a discussion.
>
>
> > **6. The proposed method is not applied to maximize validity [...] this raises concerns that the method may not be effective for enforcing structural validity. At least, the authors should discuss this point.**
>
> Our framework can handle arbitrary constraints, so we see this as an exciting opportunity. However, this paper did not focus solely on synthetic chemistry experiments, but prioritized a breadth of applications, as pointed out also positively by other reviewers.
>
> Please notice **our evaluation indeed included validity** and shows that our method **provides state-of-the-art results for generating valid and novel molecules** even if validity is not enforced as a constraint.
> In our evaluation, we outperform the prior SOTA dramatically, reporting a 48% increase in valid **and** novel generations. This increase is even more substantial when conditioning on drug-likeliness, **producing more than three times valid and novel samples as compared baselines**.
>
> To show how CCD could handle additional constraints we ran additional experiments during rebuttal in which we extended the molecule generation setting to handle an instability related constraint. Instability undermines potency and developability. The structural properties of 'three‑membered heterocycles' has been shown to exacerbate instability and reactivity [E]. For this application, we use the RDKit library [F] (the same library we use to check for chemical correctness) to flag molecules violating this structural property and implement a projection to ensure that generated molecules do not contain this structure. The projection operator consists of multiple structural editing stems. First, it expands the ring by inserting an atom so it becomes a 4- or 5-membered ring; if that isn’t possible, it then opens the ring by breaking one bond so the fragment is linear. If still flagged, it then removes the entire three-membered ring and directly reconnects the substituents.
>
> Empirically we report $0\%$ violation rate in the molecules generated with the CDD method while all the baselines report large violations.
>
> | **Model** | **Three-membered heterocycles (Viol \%)** |
> |-----------|------------------------------------------|
> | AR | $9.3 \pm 7.6$ |
> | MDLM | $22.2 \pm 2.1$ |
> | UDLM | $16.9 \pm 2.1$ |
> | CDD | $\mathbf{0.0 \pm 0.0}$ |
>
> These results show how important this technique can become for applications such as synthetic chemistry and beyond.
>
>
> > **7. The proposed method includes an optimization step at each denoising step, which suggests a potentially significant computational overhead. This limitation is not discussed in the paper.**
>
> This limitation is acknowledged and discussed in detail in the paper (limitation section, deferred to Appendix A). We also encourage the reviewer to check Appendix E, where we analyze and report the runtime. Here, we find that **CDD is orders of magnitude faster than other controllable generation methods**.
>
> Furthermore, analysis in Appendix D explores several hyperparameters that can be adjusted to control the tradeoff between quality and runtime as seen in  **Table 3**, **Figure 8**, and **Figure 9**. Also, we would like to highlight **Table 4** where we compare the overhead with the baselines and observe that CDD provides the least additional overhead as compared to other controllable generation baselines.
>
> Finally, in many settings, such as scientific tasks, generating feasible and physically plausible material largely outweigh latency considerations, given that physically unreliable content might not be even synthesizable.
>
> ---
>
> Thank you for your feedback. We hope our response addressed all your concerns; We also provided new results and noted missed items. Given your comments that these would strengthen the work, we hope that you could re-evaluate it, in light of our rebuttal.
>
>
> [A] Simple and effective masked diffusion language models
>
> [B] Simple guidance mechanisms for discrete diffusion models
>
> [C] Variational analysis
>
> [D] Constrained synthesis with projected diffusion models
>
> [E] Flavin-enabled reductive and oxidative epoxide ring opening reactions
>
> [F] G. Landrum et al. RDKit 2024.09.5 (Q3 2024) release. Zenodo, 2025
>
> [G] Grammars and reinforcement learning for molecule optimization

---

> > ### Comment · Reviewer_3U4L · 2025-08-04
> >
> > I would like to thank the authors for their responses. Some concerns have been resolved, however, several others remain:
> >
> >  1. I indeed overlooked the definition of $x'$. However, the definition of $y$ appears more than a paragraph after Eq. 5, which significantly affects readability and clarity.
> >
> > 2. The description of discrete score matching is introduced in the method section, despite not being part of your contribution. It would be more appropriate to introduce this in the related work section (I acknowledge you do state it is not novel).
> >
> > 3. The discussion of the paper’s limitations should appear in the main text. At the very least, it should be briefly mentioned with a pointer to the appendix for further detail.
> >
> > 4. Evaluation on molecules: How do you explain the discrepancy in novelty scores between your results and those reported in [8] (Schiff)?

---

> > > ### Author Response · Authors · 2025-08-04
> > >
> > > Thank you for your response! We are glad that our rebuttal has addressed the majority of your concerns, and we are more than happy to add details regarding your remaining questions.
> > >
> > > > **The definition of $\mathbf{y}$ appears more than a paragraph after Eq. 5, which significantly affects readability and clarity.**
> > >
> > > Thank you for the suggestion --  we will position the definition in line 177 immediately following the equation.
> > > To be specific, $\mathbf{y}$ is defined mathematically in Equation (5) (it's the solution to the $\arg\min$), so we will make sure this is further emphasized in the text.
> > >
> > >
> > > > **The description of discrete score matching is introduced in the method section, despite not being part of your contribution. It would be more appropriate to introduce this in the related work section.**
> > >
> > > We appreciate this suggestion and are happy to include this description in the preliminaries.
> > >
> > >
> > > > **The discussion of the paper’s limitations should appear in the main text.**
> > >
> > > This discussion was deferred to the Appendix due to space constraints, but we are happy to include it in the main text in the camera-ready version (as an additional page is provided). We recognize the importance of highlighting this, and this is indeed why we positioned it in the first section in the appendix. Thank you for your attention to this detail!
> > >
> > >
> > > > **How do you explain the discrepancy in novelty scores between your results and those reported in [8] (Schiff)?**
> > >
> > > Indeed, our novelty results match those reported by Schiff et al. While the synthetic accessibility table is not directly comparable, as we are imposing guidance on this property (which was not explored by Schiff et al.), the results reported by the baselines in our novelty experiments **exactly match those reported in the QED experiments of their paper** (see Table 5 of Schiff et al.). We hope this clarification helps!
> > >
> > > ---
> > >
> > > Thank you again for engaging with us during this discussion period! As each of your remaining concerns are formatting related and easily addressable for a camera-ready version, we hope that you would reconsider your assessment in light of this. Many thanks!

---

> ### Author Response · Authors · 2025-08-06
>
> Thank you, again, for your feedback during the response period. We believe that our response has addressed your remaining concerns, but would like to inquire whether any outstanding questions remain.
>
> Again, we appreciate the constructive feedback provided and assuredly will include these organizational revisions in a camera ready version of our paper. Thank you for your time and consideration of our work!

---

### Official Review · Reviewer_kckY · 2025-06-28

**Clarity:** 4
**Significance:** 3
**Originality:** 3
**Rating:** 4
**Confidence:** 4

**Summary:**

This paper propose a strategy for constraining the output of discrete diffusion models—at the sequence level—in a way that has otherwise been missing from traditional auto-regressive language models. It highlights how discrete diffusion models generates entire sequences-at-once, allowing for a greater level of controllability during this generative process. In practice, the proposed Constraint Discrete Diffusion (CDD) model introduces a constraint set C over the output. Rather than just applying a projection at the output, the paper shows how to sample from the projected probability distribution at every step of the diffusion procedure, selecting the $x_s$ within the feasible subdistribution. While the theory holds for convex sets C, in practice, the strategy for constraint application holds for complex non-linear constraints as well such as toxicity.  The theoretical algorithm based on Lagrange multipliers and increasing the strength of the penalty terms drives denoising to outputs that are close to the original denoised distribution but that also satisfy the given constraints. Experiments on toxicity, novelty and synthetic accessibility in molecules, and instruction all show significant improvements in constraint satisfaction in comparison to standard approaches for autoregressive models—though at some cost to generative perplexity.

**Questions:**

Questions:
1) Does the number of steps used for diffusion decoding make a significant difference in the application of constraints? Do you expect this method to work with low NFE methods such as consistency-distilled discrete diffusion models?
2) Could Constrained Discrete Diffusion be used to edit already completed generations (e.g. from the output of a AR model)?
3) Could the be a larger discussion about “constraint start ablation” and associated runtimes in the main body of the paper. Greater signposting in Appendix D could be helpful. Related, it may be interesting to explore the extent to which late-starting of constraints could be applied and the minimum number of steps required to enforce constraints. This may heavily differ between MDLM and UDLM but would make for a different level of usability.

Nits:
Including arrows for text showcasing which direction leads to an improvement in all plots (e.g. LLM-Judge Coherence in Figure 3) would help reader understanding.

**Ethical Concerns:**

["NO or VERY MINOR ethics concerns only"]

**Final Justification:**

I stand by the original score and am supportive of the acceptance of this paper. The authors were able to provider greater detail to address a number of the potential concerns (e.g. compatibility with low NFE). The authors also provided greater clarity on the importance of using UDLM for the CDD framework in order to judge the complete generation and provide constraints. The addition of these details to the final paper will be of use to the reader.

**Limitations:**

Yes

**Quality:**

3

**Strengths And Weaknesses:**

Strengths:
1) The paper is very well-written, highlighting how discrete diffusion methods open a new avenue for constrained generation and sampling.
2) The application of constraints and use of projections that minimize the distance to the original denoising distribution is theoretically well-motivated. The proof and convergence of CDD for convex sets C motivates its usage and are also well-presented in the main body of the work.
3) The experimental results are promising and well-discussed in the paper. The core experiments around nonlinear constraints on GPT-2-like language models showcase a variety of examples where constrained discrete diffusion appears an applicable solution to constrained generation.

Weaknesses:
1) The distinction between ULDM and MDLM should be discussed in greater detail, especially in the context the limitations of available and design of constraints. For example, my understanding is that the novelty and synthetic accessibility constraints applied to molecule generation in section 5.2 are only possible because the entire sequence is available at every step. This however differs from base model used for toxicity or instruction-following which utilize the masked diffusion framework. In these cases, it may be possible to enforce the constraints on partially available input (e.g. counting or lexical) but it is less clear to me how LLM-as-a-Judge or surrogate models can be used to evaluate partially decoded sequences. Please consider adding additional details and specifying why each base model was chosen in each case.
2) As the paper explicitly requires the constraints to be defined and evaluated at runtime, a number of other baselines, such as standard decoding + rejection sampling for the constraint should be considered. This may further highlight the benefits of the constrained diffusion approach, as the number of samples required could lead to drastically longer runtimes for AR models or diffusion models without explicit sampling constraints.

---

> ### Author Rebuttal · Authors · 2025-07-31
>
> We thank the reviewer for their insights, and in particular the **acknowledgement of the strong empirical results** and **well motivated theoretical support**. We provide our answers to their questions and comments.
>
> > **1. The distinction between ULDM and MDLM should be discussed in greater detail, especially in the context the limitations of available and design of constraints. [...]**
>
> Thank you for the opportunity to clarify the use of UDLM [A] and MDLM [B] and their interactions with constraints. MDLM and UDLM share the same discrete-diffusion backbone, in which they corrupt a clean sample with a schedule and then iteratively denoise to learn the target distribution. However, they use different noise kernels; MDLM uses an absorbing $\texttt{[MASK]}$ process, where once a token is unmasked it is not further updated, and UDLM uses uniform categorical noise in which any token may be revised at further states. We chose UDLM for the molecular experiments as empirically in [A] UDLM has been shown to perform better for small vocabularies, and MDLM has better performance for larger vocabularies, which covers natural language generation tasks.
>
> Crucially, *in both cases the sampler exposes a full $L \times |V|$ probability tensor at every reverse step,* and therefore **CDD always operates on the whole sequence rather than a partial generations.** The reviewer's point that some constraints are enforceable on partial generations (e.g., counting or lexical constraints) is insightful, but in many application we need global consistency. This cannot be achieved with the level of control provided with current autoregressive architectures. This is highlighted well by our molecule generation experiments, where autoregressive models report the poorest performance in generating molecules which are both novel and valid. This occurs as the models often reach intermediate states where no valid and novel continuations remain. Consequently, they become "stuck," and either reproduce known samples (violating the novelty constraint) or generate invalid structures (violating molecule validity).
>
> Even in Section 5.3, where it may be possible to enforce constraints on partial generations, we find that there are distinct benefits to imposing the constraints in a non-autoregressive paradigm, as evidenced by better PPL and LLM-as-a-Judge scores. Specifically, lexical constraints benefitted from non-autoregressive generation as tokens both preceding and following the constrained positions can be simultaneously adjusted, a flexibility unavailable to strictly autoregressive models.
>
> As a final note, the LLM-as-a-Judge is evaluation-only on the **post-generation**. During sampling we use a differentiable surrogate models (or other differentiable constraint functions) to evaluate the *entire sequences*, not partial generations. After applying the Gumbel SoftMax, we then embed these distributions directly into autoregressive LLM-based surrogates (using the same tokenizer as the base model), creating a continuous, fully differentiable computational graph. In scenarios involving hard constraints, we leverage hand-coded rule-based constraint functions, enabling us to avoid relaxation and embedding, and directly evaluate the argmax-decoded sequences. More details on these implementations are provided in Appendix C.
>
> We thank you for bringing up this point and will add a concise subsection clarifying these points and explicitly justify the base model choice per task.
>
>
> > **2. Does the number of steps used for diffusion decoding make a significant difference in the application of constraints? Do you expect this method to work with low NFE methods such as consistency-distilled discrete diffusion models?**
>
> While the number of denoising steps effects efficiency and sample quality, it does not prevent CDD from enforcing constraints. As the projection operates directly on token‑level probability outputs, **CDD is fully compatible with low‑NFE samplers**, including consistency‑distilled discrete diffusion models, without architectural changes.
>
> The performance on such methods could be extrapolated from our analysis in Appendix D, where *we analyze the effects of reducing the number of projections* (by taking more diffusion steps between each projection). The only difference would be that low‑NFE samplers would take a single large step between projections, whereas in our analysis many small steps were taken between projections. We observed that performance only degraded slightly when a higher number of steps occurred between projections, and, based on this result, we believe that this suggestion could be a promising area to explore in future work as consistency-distilled discrete diffusion models become more readily available. Thank you for pointing it out!
>
>
> > **3. As the paper explicitly requires the constraints to be defined and evaluated at runtime, a number of other baselines, such as standard decoding + rejection sampling for the constraint should be considered. This may further highlight the benefits of the constrained diffusion approach, as the number of samples required could lead to drastically longer runtimes for AR models or diffusion models without explicit sampling constraints.**
>
> Indeed, **rejection sampling is implicitly included as one of our baselines**. In all experiments we report the percentage of samples which violate the constraint set. These samples can be treated as those which would have been 'rejected'. We'd be happy to highlight this point in the main text.
>
>
> > **4. Could Constrained Discrete Diffusion be used to edit already completed generations (e.g. from the output of a AR model)?**
>
> This is an interesting suggestion and could be an intriguing application of CDD. In fact, our research team has already been exploring this idea for continuous diffusion models; in such settings, **we found that there are some significant benefits that can be realized**, which very likely extend to discrete settings as well. In short, CDD could be applied in this way by initializing the input sequence with an existing sequence. Deviation from the original sequence could be controlled by tuning the masking schedule (and how many diffusion steps to take). However, if the original output is quite far from the constraint set (e.g., imposing toxicity constraints on a very harsh sequence), this may result in a largely changed sequence. Thank you again for this suggestion!
>
>
> > **5. Could the be a larger discussion about “constraint start ablation” and associated runtimes in the main body of the paper. Greater signposting in Appendix D could be helpful. Related, it may be interesting to explore the extent to which late-starting of constraints could be applied and the minimum number of steps required to enforce constraints. This may heavily differ between MDLM and UDLM but would make for a different level of usability.**
>
> We appreciate this suggestion. It adds to the clarity and we will implement these suggestions on the updated manuscript. To the latter point, note also that *the question of late-starting of constraints and the minimum number of steps required to enforce constraints is a result we have characterized theoretically* (Theorem 4.1). We are happy to pair this explanation with the empirical analysis conducted in Appendix D in the updated version of our paper.
>
>
> ---
>
> We thank you for your suggestion and are happy to include your suggestions in the final version and to engage further if there are any additional questions. Thank you!
>
>
> [A] Schiff, Yair, et al. "Simple guidance mechanisms for discrete diffusion models." arXiv preprint arXiv:2412.10193 (2024).
>
> [B] Sahoo, Subham, et al. "Simple and effective masked diffusion language models." Advances in Neural Information Processing Systems 37 (2024): 130136-130184.

---

> > ### Comment · Reviewer_kckY · 2025-08-05
> > **Discussion**
> >
> > We thank the authors for the rebuttal and the additional details. I have read through the reviews and rebuttal comments and maintain my support for the paper.
> >
> > From the rebuttal,
> >
> > (1) thanks for additional clarity on the role of UDLM and MDLM. It is interesting how CDD is able to apply constraints by applying itself to the fully generated sequence. To be clear, it seems to me that CDD is most compatible with UDLM but not standard MDLM approaches as the factorization of the joint (even using the arbitrary ordering) could still lead to `stuck' generations.
> >
> > (2/5) Thanks for pointing out this detail in the appendix. Highlighting it in the main text in the final version of the paper would help signpost this result.
> >
> > (4) sounds great, thanks for the update. It does seem possible that using SDEdit like methods for applying constraints or various sorts of control on input sequences can be a viable approach to diffusion.

---

> > > ### Author Response · Authors · 2025-08-06
> > >
> > > We thank you for your response and continued support of the paper.
> > >
> > > > **(1) thanks for additional clarity on the role of UDLM and MDLM. It is interesting how CDD is able to apply constraints by applying itself to the fully generated sequence. To be clear, it seems to me that CDD is most compatible with UDLM but not standard MDLM approaches as the factorization of the joint (even using the arbitrary ordering) could still lead to `stuck' generations**
> > >
> > > We appreciate the opportunity to further clarify! To make sure we understand the underlying question: are you asking whether the masking schedule by MDLM interferes with the application of our constraints?
> > >
> > > If this is the case, first, although MDLM unmasking is irreversible, note that CDD applies its projections to the full-sequence probability distribution, *including over tokens that have already been unmasked*. Hence, even when tokens are unmasked, the projection operator is always able to modify the probability distributions to restore feasibility.
> > > \
> > > Second, it is important to note that projections are applied **not** only when the sequence is fully generated, but after early denoising steps where all tokens are still masked. These early and continuous projections ensure that the generations need minimal corrections later in the generation, mitigating the risk of generations getting ``stuck'' as feasible, but unrealistic, generations.
> > >
> > > Finally, our masking schedule has indeed been tuned to account for this issue. We also note that more recent discrete diffusion works (e.g., [1]) have proposed adaptations to the schedule that encourage later-stage masking. Doing so, masked diffusion models can be effectively tuned, especially in cases where the suggested issue may arise!
> > > \
> > > While we didn't observe generations getting ``stuck'', these masking schedules can certainly be leveraged in our framework.
> > >
> > >
> > > > **(2/5) Thanks for pointing out this detail in the appendix. Highlighting it in the main text in the final version of the paper would help signpost this result.**
> > >
> > > We are happy to highlight these points in the main text in the final version. We further thank you for raising these points!
> > >
> > > > **(4) sounds great, thanks for the update. It does seem possible that using SDEdit like methods for applying constraints or various sorts of control on input sequences can be a viable approach to diffusion.**
> > >
> > > Fantastic. We agree this is a compelling area to explore in future work! Thank you for the comment!
> > >
> > > ---
> > > Thank you for your response and let us know if there is any further clarification we could provide.
> > >
> > >
> > > [1] Shi, Jiaxin, et al. "Simplified and generalized masked diffusion for discrete data." Advances in neural information processing systems 37 (2024): 103131-103167.

---

### Official Review · Reviewer_v7jP · 2025-07-02

**Clarity:** 2
**Significance:** 2
**Originality:** 2
**Rating:** 3
**Confidence:** 2

**Summary:**

The paper introduces Constrained Discrete Diffusion, a framework that integrates discrete diffusion models with differentiable constraint optimization to enforce sequence-level constraints during generation. CDD leverages the parallel denoising process of diffusion models to impose constraints iteratively. The key innovation is a projection operator that minimizes KL divergence between the denoised distribution and a feasible set, ensuring constraint satisfaction while preserving sample quality. The method is evaluated on three tasks: toxicity-controlled text generation, property-constrained molecule design, and instruction-following text completion.

**Questions:**

- Since $\mathbf{C}$ represents the constraint set and $\mathbf{y}$ denotes the projected probability distributions in Eq.5, the meaning of $\arg\max(\mathbf{y}) \in \mathbf{C}$ is quite confusing. What does it mean in the task of natural language toxicity mitigation?
- Surrogate models are trained on clean data but applied to noisy intermediate samples during diffusion. How does this affect their reliability?
- While Table 4 shows CDD’s runtime overhead, how does the frequency of projection steps impact sample quality?

**Ethical Concerns:**

["NO or VERY MINOR ethics concerns only"]

**Final Justification:**

The authors have provided a explanation of the projection formulation in Eq.5 and committed to improving its exposition in the revision. Despite clarifications, Section 4 as currently written is still opaque, and the key technical steps require more intuitive explanation to be accessible to a broader audience.
The evaluation remains somewhat narrow; the generalizability to substantially different constraints or settings is still mostly argued rather than demonstrated empirically.
The paper is technically sound and makes a novel contribution. However, the clarity and breadth of empirical evaluation limit its accessibility and impact in its current form.

**Limitations:**

yes

**Paper Formatting Concerns:**

No major formatting issues were found. The paper follows the guidelines.

**Quality:**

3

**Strengths And Weaknesses:**

### Strengths
- The paper is technically sound, with rigorous theoretical grounding (e.g., convergence guarantees in Theorem 4.1) and experiments validating the method’s effectiveness. The ablation studies and runtime analysis further strengthen the empirical claims.
- The training-free nature of the approach is a practical advantage. CDD addresses a critical limitation of autoregressive models by enabling hard constraint satisfaction, which is valuable for safety-critical applications (e.g., toxicity mitigation) and scientific tasks (e.g., molecule design).
- The integration of Lagrangian optimization with discrete diffusion is novel.

### Weaknesses
- Designing task-specific projection operators adds overhead, limiting plug-and-play usability. The paper could better discuss generalizability to unseen constraints.
- Since some tasks require training a surrogate model, my concern is whether a model trained on noiseless data can accurately offer effective guidance when the input is incompletely denoised intermediates during the sampling process.
- While the method is training-free, the projection steps introduce runtime overhead (Table 4). The trade-off between constraint adherence and speed could be analyzed further.

---

> ### Author Rebuttal · Authors · 2025-07-31
>
> We thank the reviewer for their comments, and, in particular, the **acknowledgement of the rigorous theoretical grounding** and **novelty of our work**. Below we provide our answers to their questions.
>
> > **1. Since $\textbf{C}$ represents the constraint set and $\textbf{y}$ denotes the projected probability distributions in Eq.5, the meaning of $ \arg\max(\textbf{y}) \in \textbf{C}$ is quite confusing. What does it mean in the task of natural language toxicity mitigation?**
>
> In Equation~(5), the constraint is placed not on the probability tensor itself but on the decoded sequence $\textbf{y}^{\star}=\arg\max\bigl(\textbf{y}\bigr)$ where $\textbf{y}\in\mathbb{R}^{L\times |V|}$ is the projected probability distribution.
> As greedy decoding picks the highest–probability token at each position, the condition $\arg\max(\textbf{y})\in\textbf{C}$ means the decoded sequence must be feasible. Specifically for toxicity control $\mathbf{C}=\lbrace\mathbf{y}\in V^{L} : g(\mathbf{y}^{\ast})\le \tau \rbrace,$ where $g(\cdot)$ is a differentiable toxicity surrogate and $\tau\in[0,1]$ is the threshold. Therefore, in the task of natural language toxicity mitigation, $\arg\max(\textbf{y})\in\textbf{C}$ is the set of sequences which are less toxic than $\tau$ as score by the surrogate model. Since $\arg\max$ is non-differentiable, the projection optimizes an augmented–Lagrangian objective using a Gumbel Softmax relaxation $\tilde{\phi}(\textbf{y})$ and penalizes violations of $g\bigl(\tilde{\phi}(\textbf{y})\bigr)\le \tau$.
> *In the reported benchmarks, this procedure yielded $0\%$ violations at the tested thresholds without retraining the base model!*
>
>
> > **2. Surrogate models are trained on clean data but applied to noisy intermediate samples during diffusion. How does this affect their reliability?**
>
> This is an interesting question, which is closely connected to ongoing discourse within the context of classifier-based guidance methods [A-C]. Specifically, recent works exploring **"training-free"** guidance methods have analyzed how pretrained classifiers, trained exclusively on clean data, could be directly adopted for diffusion model guidance without explicit finetuning on noisy samples. The trade-off here is well established: employing pretrained models directly on noisy inputs significantly reduces implementation complexity, albeit at the cost of slightly reduced performance due to distribution shift. *Our adoption of surrogates trained exclusively on clean data can thus be understood within this well-established trade-off;* indeed, *we observed strong empirical results even without noisy-domain finetuning*.
>
> Next, it is important to clarify what "reliability" means in this context. We interpret this as the accuracy with which the surrogate models the constraint set. Let us stress that *the combinatorial set of feasible generations defined by our surrogate remains unchanged regardless of the input noise*. As the surrogate model defines a closed set of allowable generations, **the noise level of the measured sequences does not compromise the integrity of the surrogate's prediction of set membership**. At worst, the noisy data may lead to slightly less accurate gradients (as documented by training-free guidance literature), introducing slightly longer runtime for the convergence of the ALM projection. However, as we have theoretically shown (Theorem 4.1), CDD effectively converges towards the training data distribution while simultaneously converging to the constraint set.
>
>
>
> > **3. While Table 4 shows CDD’s runtime overhead, how does the frequency of projection steps impact sample quality?**
>
> Thank you for bringing up this important point. **Indeed, we examine this in Appendix D, where we ablate the projection frequency over various frequencies $\{1,5,10,20,50,100\}$ and report the average perplexity and runtime in Figure 8.** We also include sensitivity studies for Augmented Lagrangian hyperparameters in Table 3 and for the projection start time in Figure 9. Notably, across all settings, constraints remain fully satisfied (0% violations), underscoring the robustness of our method. In practice this method comes with practical knobs for practitioners which allows users to control compute, speed, and generation quality while preserving feasibility.
>
>
> > **4. Designing task-specific projection operators adds overhead, limiting plug-and-play usability. The paper could better discuss generalizability to unseen constraints.**
>
> We appreciate the reviewer highlighting this point. Indeed, this concern closely relates to the earlier discussion in Response (2) about training-free guidance. In that context, we noted that our approach explicitly avoids reliance on surrogate models pretrained on noisy diffusion outputs. Consequently, **training a surrogate model for arbitrary, unseen constraint sets is relatively straightforward**, requiring only standard supervised training procedures without specialized diffusion-domain fine-tuning. This is consistent with prior work on attribute-controlled diffusion using surrogate models where constraint-specific loss functions or guidance models are similarly required [D,E]. Thus, the method naturally generalizes to a wide variety of constraints with minimal overhead.
>
> Furthermore, we have also demonstrated that the approach generalizes to other representations of constraint (e.g., symbolic, black-box, neural surrogate), which are treated as **modular inputs to the optimization procedure**. For a given setting, it is *only necessary to devise a differentiable constraint function* -- which, again, can take many forms. Therefore, rather than requiring "task-specific projection operators," our method only necessitates defining differentiable "constraint violation functions." We emphasize that defining these constraint functions has a long tradition in optimization and should not be seen as a barrier to generalizability.
>
> We would be glad to explicitly incorporate additional explanation into the paper, emphasizing the distinction between task-specific projection operators (which our method avoids) and differentiable constraint representations.
>
>
> ---
>
> Thank you for your detailed review! We hope that our response addressed all of your questions. We are also happy to provide any other requested details.
>
>
>
> [A] Sadat, Seyedmorteza, et al. "No training, no problem: Rethinking classifier-free guidance for diffusion models." arXiv preprint arXiv:2407.02687 (2024).
>
> [B] Ye, Haotian, et al. "Tfg: Unified training-free guidance for diffusion models." Advances in Neural Information Processing Systems 37 (2024): 22370-22417.
>
> [C] Shen, Yifei, et al. "Understanding and improving training-free loss-based diffusion guidance." Advances in Neural Information Processing Systems 37 (2024): 108974-109002.
>
> [D] Schiff, Yair, et al. "Simple guidance mechanisms for discrete diffusion models." arXiv preprint arXiv:2412.10193 (2024).
>
> [E] Gruver, Nate, et al. "Protein design with guided discrete diffusion." Advances in neural information processing systems 36 (2023): 12489-12517.

---

> > ### Comment · Reviewer_v7jP · 2025-08-01
> >
> > Thank you for the detailed response. Most of my concerns have been addressed satisfactorily. I will maintain my score as I believe two aspects could benefit from further clarification or empirical support:
> >
> > 1. While the rebuttal helps clarify the meaning of the projection formulation, the current exposition in the Section 4 (especially Equation 5) remains somewhat opaque.
> > 2. Although the authors cite relevant training-free guidance literature and provide a theoretical rationale, I believe the paper would be strengthened by including ablation studies that more directly measure the impact of input noise (e.g., different noise levels) on surrogate effectiveness.

---

> > > ### Author Response · Authors · 2025-08-01
> > >
> > > Thank you for your quick response and your willingness to engage in discussion with us during this period. We are happy to hear that our response has addressed the majority of your concerns and are happy to provide further clarification on the points that remain.
> > >
> > > > **While the rebuttal helps clarify the meaning of the projection formulation, the current exposition in the Section 4 (especially Equation 5) remains somewhat opaque.**
> > >
> > > We would be more than happy to integrate the points discussed in our rebuttal into Section 4. As you have acknowledged that the description we provided has clarified the projection formulation (Equation 5), we believe the inclusion of these points should resolve that concern.
> > >
> > >
> > > > **Although the authors cite relevant training-free guidance literature and provide a theoretical rationale, I believe the paper would be strengthened by including ablation studies that more directly measure the impact of input noise (e.g., different noise levels) on surrogate effectiveness.**
> > >
> > > As you mention, our rebuttal included theoretical rationale as to why it is unnecessary to train the surrogate on noisy data. However, we are happy to provide empirical rationale as well. First, let us highlight that in both experimental settings where surrogate models are leveraged, toxicity mitigation and molecule generation with synthetic accessibility constraints, **we achieve perfect constraint satisfaction -- not just at the final state but at all intermediate states as well**. Our augmented Lagrangian projection *always converges in the experimental settings we tested, regardless of the noise level*. We would be happy to highlight this result further in the paper, as we believe this is clear empirical evidence of the surrogate's effectiveness.
> > >
> > > Furthermore, we indeed tried training a surrogate using noisy data during preliminary development of the method. Empirically, we found no significant difference in performance between this surrogate model and one trained exclusively on clean data. The sole discrepancy we observed was that it took much longer for the surrogate trained on noisy data to converge. Hence, we proceeded with our experiments using a training-free approach outlined in the paper.
> > >
> > > Combining these empirical justifications with the theoretical rationale that we previously provided, we contest that our responses have provided compelling evidence that the surrogate models employed in our study are effective regardless of the noise level.
> > >
> > > ---
> > >
> > > We sincerely hope these responses have addressed your remaining concerns, and we would greatly appreciate if you would reconsider your evaluation in light of these clarifications. Many thanks!

---

> > > > ### Comment · Reviewer_v7jP · 2025-08-06
> > > >
> > > > Thank you for your follow-up and the additional clarification you provided in your response. While my overall assessment remains unchanged after considering your responses to all reviewers and the broader discussion, I will continue to actively follow up on the discussion with you and the other reviewers.

---

> > > > > ### Author Response · Authors · 2025-08-06
> > > > >
> > > > > Thank you for letting us know that all of your concerns are clarified.
> > > > > \
> > > > > We appreciate your continued engagement and are ready to clarify any additional point that may help the discussion move forward. Rest assured that we will address all comments discussed (e.g., exposition of the mathematical concepts pointed out, the discussion of training-free literature) will be addressed in the camera ready version.
> > > > > \
> > > > > We appreciate your time and willingness to reconsider your evaluation in light of these clarifications.
> > > > >
> > > > > Many thanks!

---

### Official Review · Reviewer_sK3h · 2025-07-02

**Clarity:** 3
**Significance:** 2
**Originality:** 1
**Rating:** 4
**Confidence:** 3

**Summary:**

This paper proposes a new method for generating discrete data that strictly satisfies constraints on the sequence-level. The proposed algorithm, based on discrete diffusion, uses a gradient-based approach to project the denoised sequence onto the closest sequence that satisfies the set of constraints.

**Questions:**

1. In what respect does your paper differ substantially from [1]? It also uses an inner loop with gradient-based optimisation to optimise for a specific objective, which also exactly includes a KL divergence term and a reward term (here, a constraint term). Please point out clear differences. I could be mistaken in my understanding of your method.
2. While the method works well, it requires a great deal of additional runtime (which, I understand, can be a cost that is absolutely fine to pay, in some contexts). Do you believe your method could be sped up in any simple way?
3. Have you tried methods such as ReinMax [2], that allow for differentiable discrete sampling instead of Gumbel softmax?
4. Have you attempted your method in combination with other SOTA sampling methods, such as P2 [3]?

**Ethical Concerns:**

["NO or VERY MINOR ethics concerns only"]

**Final Justification:**

While the paper presents little originality in its method, it sets a new baseline for constrained generation using discrete diffusion. Discrete diffusion being a very active topic in the community, it will most certainly be referred to in further works to compare with arguably a relatively bruteforce method, which admittedly does empirically work.

**Limitations:**

I am concerned about the novelty of the work, primarily.

References
---
[1] Protein Design with Guided Discrete Diffusion (2023). Nate Gruver, Samuel Stanton, Nathan C. Frey, Tim G. J. Rudner, Isidro Hotzel, Julien Lafrance-Vanasse, Arvind Rajpal, Kyunghyun Cho, Andrew Gordon Wilson.

[2] Bridging Discrete and Backpropagation: Straight-Through and Beyond (2023). Liyuan Liu, Chengyu Dong, Xiaodong Liu, Bin Yu, Jianfeng Gao.

[3] Path Planning for Masked Diffusion Model Sampling (2025). Fred Zhangzhi Peng, Zachary Bezemek, Sawan Patel, Jarrid Rector-Brooks, Sherwood Yao, Avishek Joey Bose, Alexander Tong, Pranam Chatterjee.

**Paper Formatting Concerns:**

-

**Quality:**

2

**Strengths And Weaknesses:**

Strengths:
1. The paper’s presentation is clear and rather simple. The goals and methods are understandable.
2. The empirical evaluation seems rigorous overall and is extensive.
3. The method empirically achieves the desired goal of no violation whatsoever of the desired constraints.

Weaknesses:
1. While overall the paper is not too poorly written, section 4 should be broken down into subsections, as it is currently quite hard to read, in terms of structure.
2. The authors make no mention of [1], while it is an extremely close method to the one they propose. In [1], authors also propose a gradient-based approach to optimise for a reward while generating a sequence with discrete diffusion. As a matter of fact, I would like the authors to show the novelties their work offers when compared to [1]; it is my biggest concern about this work.
3. Theoretically, the paper is relatively weak. Moreover, while the proof of Theorem 4.1 seems relatively sound, I fail to follow a few parts which could, I believe, use a bit of clarification. Namely, “ $\forall x, y$ in *the* neighbourhood of $C$ ”: what does this mean? which neighbourhood? (Maybe in “a” neighbourhood?) What is $\beta$-prox regularity? These are just a few examples, and the paper contains a few (minor) typos here and there. I am not so concerned by this point, as the paper evidently is more oriented towards experiments.

Overall, I think the paper is of good quality, but I am truly concerned about its novelty. I would be happy to reevaluate my opinion on that matter, should the authors provide good evidence.

---

> ### Author Rebuttal · Authors · 2025-07-31
>
> Thank you for the helpful review, and, in particular, for the acknowledgement of the *rigorous and extensive empirical evaluation and performance*. We address below the outstanding questions.
>
> > **1. In what respect does your paper differ substantially from NOS?**
>
> Our method aims to solve a **different mathematical problem** and has a **different behavior and guarantees** when compared to NOS. Specifically, while NOS appends a gradient term to the denoising step for soft guidance, our method implements an augmented Lagragian projection operator that enforces **all samples to lie in the constraint feasibility set $\textbf{C}$**.
> Within the inner loop, our method solves the projection problem: find the nearest point to the original sample that lies in the constraint set $\textbf{C}$. Differently from NOS, the KL term here is used as distance measure in the projection objective: it keeps the new sample as close as possible to the model’s current score distribution. Another novelty of our framework is the theoretical ground provided: Feasibility (as convergence to a constrained set) is reported in Theorem 4.1. Additionally, empirically, CDD shows zero constraint violations in every experiment tested.
>
>
> In contrast, in NOS the KL term acts a regularizer between the current and original logits with the $\lambda$ term regulating the trade‑off between distance and reward. The guidance is encoded only through this scalar reward and depends on tuning $\lambda$. It also provides no convergence guarantees. Qualitatively this technique is largely outperformed by our method. For example, [B] assesses the NOS guidance scheme for the same molecule domain we study in this paper, adapting it to MDLM and UDLM. Below we provide a side-by-side comparison for the novelty setting:
>
> | **Model**  | **Valid & Novel** | **QED** | **Viol (%)** |
> |------------|-------------------|---------|--------------|
> | UDLM       |         345       |   0.46  |    61.45     |
> | UDLM + NOS |         159       |   0.47  |    71.01     |
> | UDLM + CDD |         511       |   0.45  |     0.00     |
>
>
> Note how adding NOS to UDML (second row) actually increases the violations by ~10\% with respect to its baseline while adding our constrained projection method (last row) reduces these violations to 0.
> Hence, we hold that our comparison in Figure 4 effectively illustrates significant improvement over the state-of-the-art, illustrating for the first time such drastic constraint reductions in these challenging domains.
>
> We appreciate the question and will make sure to explain these differences in an updated Related Work section.
>
>
> > **2. Theoretically, the paper is relatively weak. Moreover, while the proof of Theorem 4.1 seems relatively sound, I fail to follow a few parts which could, I believe, use a bit of clarification. Namely, "$\forall x, y$ in the neighbourhood of $\textbf{C}$ ": what does this mean? which neighbourhood? (Maybe in “a” neighbourhood?) What is $\beta$-prox regularity?**
>
> First, let us highlight that to the best of our knowledge, CDD provides the first framework for discrete sequence modeling which incorporates theoretical guarantees for *non-asymptotic feasibility*, while also providing a formulation which enables modeling of *arbitrary constraint sets*.
>
> As specified in Theorem 4.1, we adopt the notion of $\beta$-prox regularity as defined by [A] (Def. 13.27). This is a classical tool in convex optimization theory which implies that for each viable point and normal direction, small perturbations still project uniquely and smoothly back to $\textbf{C}$. The "neighborhood" referenced is defined in this sense. It indicates the region in which the prox‑regularity inequalities hold. The fact that our framework, relying on a principled methodology, allows us to derive theoretical guarantees should be seen as a strengths in our view.
> We appreciate your suggestion and will explicitly present this definitions in the final version of the paper.
>
>
> > **3. While the method works well, it requires a great deal of additional runtime (which, I understand, can be a cost that is absolutely fine to pay, in some contexts). Do you believe your method could be sped up in any simple way?**
>
> Yes, there are ways to accelerate this method, although they lie outside the scope of our initial investigation.
> First, however, let us notice that, when compared to other controllable generation baselines, CDD provides the least additional overhead (see Table 4).
>
> Our analysis in Appendix D explores a few of the "knobs" that can be adjusted when controlling the tradeoff between quality and runtime. Most notably, by adjusting the projection frequency and the timestep at which projection steps begins, one can control sampling speed and quality. Our results in Table 3, Figure 8, and Figure 9 show that starting to projection at later steps, alongside less frequent projections, results in faster sampling times while still satisfying the desired constraints.
>
> Finally, as you mention, in high-assured context (e.g., scientific tasks) certificates of constraint satisfaction outweigh latency considerations.
> Indeed, providing outputs that violate critical constraints and physical principles make them unsuitable for production and testing, potentially slowing down, instead of accelerating, the AI-driven discovery pipeline.
>
>
> > **4. section 4 should be broken down into subsections**
>
> Thank you for this suggestion. We agree and will update the final version. In particular, we will include subsections covering the following concepts:
>
> 1. Projected Diffusion Sampling
> 2. Differentiable Projection
> 3. Augmented-Lagrangian Projection
> 4. Theoretical justification
>
> Thanks again!
>
>
> > **5. Have you tried methods such as ReinMax [C], that allow for differentiable discrete sampling instead of Gumbel softmax?**
>
> This is an interesting suggestion, and we certainly will consider this as a direction for future work. However, a few concerns were uncovered by our early investigation during the rebuttal phase.
> First, ReinMax depends on multiple forward-backward passes which introduce further overhead; this will be *especially challenging for the high-dimensional problems* (e.g., we use a ~60K vocabulary). In fact [C] only tests on toy problems (with *independent 128 bernulli variables)*, thus there a significant concern as to the scaling of this approach.
> Second, we suspect that ReinMax's reliance on Taylor approximations *may introduce bias which could impact the projection dynamics of CDD*. That said, we look forward to test it in our future work.
>
>
> > **6. Have you attempted your method in combination with other SOTA sampling methods, such as P2 [D]?**
>
> We appreciate the suggestion, which may constitute an interesting follow-up study. For this paper, we have not provided such a comparison for a few primary reasons. First, our experimental analysis is designed to ablate the performance of *our proposed method*; to accurately assess the efficacy of projected discrete diffusion sampling processes, we validate this approach in isolation, rather than introducing variance through a composite approach. As **such variations of traditional sampling techniques are yet to be adopted as de facto sampling schemes**, it is most practical to assess our approach using the most common sampling processes.
>
> Second, while CDD and P2 are not fundamentally incompatible, as [D] reorders the denoising trajectory while CDD enforces feasibility by projecting each intermediate distribution (orthogonal objectives), there are several practical limitations of P2 that make us hesitant to pursue this particular direction. One concern is that the additional overhead that is induced by P2 will compound with the inherent complexity introduced when certifying constraint satisfaction. Another challenge is that P2 may not generalize well to the domains that we explore in this paper, as the robustness of different planners has yet to be fully tested under domain shift. Hence, while we believe this certainly could be an interesting direction to explore, the work would likely fit the scope of an independent investigation.
>
>
> ---
>
> Thank you for your detailed review! We hope that our response addressed all of your questions. We are also happy to provide any other requested details.
>
>
> [A] Rockafellar, R. Tyrrell, and Roger JB Wets. Variational analysis. Berlin, Heidelberg: Springer Berlin Heidelberg, 1998.
>
> [B] Schiff, Yair, et al. "Simple guidance mechanisms for discrete diffusion models." arXiv preprint arXiv:2412.10193 (2024).
>
> [C] Bridging Discrete and Backpropagation: Straight-Through and Beyond (2023). Liyuan Liu, Chengyu Dong, Xiaodong Liu, Bin Yu, Jianfeng Gao.
>
> [D] Path Planning for Masked Diffusion Model Sampling (2025). Fred Zhangzhi Peng, Zachary Bezemek, Sawan Patel, Jarrid Rector-Brooks, Sherwood Yao, Avishek Joey Bose, Alexander Tong, Pranam Chatterjee.

---

> > ### Comment · Reviewer_sK3h · 2025-08-04
> >
> > I would like to thank the authors for their time spent answering my questions.
> >
> > Agreed on questions 2, 4, 5 and 6. Consider them sorted. Thank you again.
> >
> > Question 1. In your work, "the KL term here is used as distance measure in the projection objective: it keeps the new sample as close as possible to the model’s current score distribution."
> >
> > In NOS, "NOS the KL term acts a regularizer between the current and original logits with the $\lambda$ term regulating the trade‑off between distance and reward."
> >
> > Perhaps, something eludes to me, and I would be happy to change my mind on this, but I do not see how this is different. In both cases, the KL creates the tension between optimising for the objective (for them, increasing a reward; for you, satisfying a softened constraint) and remaining close to the original distribution.
> >
> > I do understand that the problems at hand are different; I never claimed the contrary. However, the difference is not fundamental: in NOS, a continuous reward is optimised (to increase it); in your case, although the original problem is that of satisfying certain "hard"/discrete constraints, you are actually minimising another reward through your $\Delta g$. So you are in fact decreasing a reward in your method as well. Similarly, you do need differentiability in the underlying constraint evaluation as you point out, as your method is _also_ gradient-based.
> >
> > Finally, the algorithm is _very_ similar – the main difference residing in your added inner loop. I understand that, empirically, your method works and perhaps NOS not, although it may be made functional with some adjustments (but we can ignore that point). I am only questioning the conceptual novelty of the work with this question; I, of course, acknowledge the good empirical evidence that shows that the method never violates the said constraints. Happy to review my position. Perhaps I am misunderstanding something.
> >
> > Question 3. Thank you. I understand that you can lower the weight on the constraints and such, but I was just curious about whether you had any ideas of directions for improvement for speed. This point is not crucial for this current paper, in my opinion, but it is obviously of interest.

---

> > > ### Author Response · Authors · 2025-08-05
> > >
> > > Thank you again for engaging with us during this discussion period.
> > >
> > > > **In what respect does your paper differ substantially from NOS?**
> > >
> > > There are two key points that we'd like to emphasize to answer this question:
> > >
> > > 1. First, please note that *NOS takes only a single gradient step after each denoising update*. This makes NOS similar in spirit to classifier-based guidance, that are popular for continuous diffusion models. In contrast, *our proposed method solves a constrained optimization problem (the constrained projection) at each step*. This means that we run a primal-dual optimization algorithm within each diffusion step. In this paper, the induced primal-dual problem is solved via a first-order method, and that is the reason why you notice *a composite number of gradient steps to find the nearest feasible point*.
> > >
> > > 2. Next, In NOS the KL term and reward function are both tunable slack penalties that are manually set. This is an important difference since the "weighted" constrained penalty terms in our method comes from the primal-dual formulation. We seek to find the strongest Lagrangian Dual relaxation:
> > > $$
> > >   \\;\\;\\;\\;\\;\\;\\;\\;  \arg\max_{\lambda, \mu} \Bigl( \arg\min_{\mathbf y} \bigl( \mathcal{L}_{\mathrm{ALM}} (\mathbf{y}, \mathcal{U}\(\mathbf{x}_t); \lambda, \mu)  \bigr)   \Bigr).
> > > $$
> > > where $\lambda$ and $\mu$ represent the Lagrangian multipliers. This, in turn, is solved via a dual ascent-inspired method, which is the reason why you see an inner loop (that optimizes the "primal" -- the objective of the projection in this case), and an outer loop (that updates the Lagrangian dual variables).
> > >
> > >
> > > The above comes from duality theory, and it ensures to find the best (when strong duality hold) Lagrange multipliers for the problem; this is very different from the "hand-tuned" weighting parameters present in NOS.
> > >
> > > We understand that there are a lot of details here, but we hope this discussion has better clarified how our approach and NOS differ. While we in no way intend to downplay the significance of NOS, we solely intend to emphasize the distinct differences between their perspective and ours. Hence, while there are certain metrics and notation that is common between our approach and NOS, this is because we are operating on the same types of problems (e.g., measuring deviation between probability distributions leads to the use of KL divergence) and not because we are approaching these problems in similar ways.
> > >
> > >
> > > > **I understand that you can lower the weight on the constraints and such, but I was just curious about whether you had any ideas of directions for improvement for speed.**
> > >
> > > Thanks for clarifying. One other direction we are currently working on, albeit in a different context, is "warm-starting" the Augmented Lagrangian method with a previous solution. When this warm-start is available (e.g., by saving the solution from the feasible previous projection step it can substantially reduce runtime. Of course, a note to take into account here is related to the constraint geometry -- when the set is loosely convex (e.g., Section 5.3) [A], this approach works very well, but in practice this approach tends to produce positive results even in the case of nonconvex constraints.
> > >
> > > ---
> > >
> > > Thank you again for engaging with us during this discussion period! We appreciate your constructive comments and hope that these clarifications have addressed all of your questions. Again, we are happy to provide any additional details.
> > >
> > > [A] Wu, B. & Bemporad, A. (2022). A Simple and Fast Coordinate-Descent Augmented-Lagrangian Solver for Model Predictive Control.

---

> > > > ### Comment · Reviewer_sK3h · 2025-08-06
> > > >
> > > > 1. NOS does not take a single gradient step after each denoising step. See algorithm 2 in the paper. It is true, however, that the number of iterations is fixed in advance by a constant. (Seeing it this way, I know that you indeed have then two inner loops for each denoising step: the outer one on $\Delta(g)$, and the inner one, also fixed by a constant, `max_inner_iter`.)
> > > >
> > > > 2. I understand, however, that your formulation does solve a more grounded problem, and that the algorithm also updates the mentioned penalty and slack coefficients algorithmically as you mentioned.
> > > >
> > > > I do not think that we disagree that the method has differences; I have noticed them as well. My main reproach is that the differences are few, and the paper's improvement _on the methods' side_ is arguably incremental, in my estimation. I do understand that your method is the first to empirically enable, in the chosen settings, generating samples that never violate required constraints.
> > > >
> > > > To conclude, while I believe that the originality is low, I think the method does introduce a new and even first baseline for these constrained discrete diffusion methods. I shall raise my score to 4.
> > > >
> > > > Thank you for having engaged in this rebuttal.

---

> > > > > ### Author Response · Authors · 2025-08-06
> > > > >
> > > > > Once again, thank you for the continued engagement during the discussion period!
> > > > >
> > > > > Indeed, Algorithm 2 of NOS performs a bounded number of iterations; Thank you for pointing this out. As you noted, the theoretical and empirical differences between our method and NOS are significant within contained generation literature, and we appreciate your acknowledgement of this. As an outcome of this discussion, we intend to add details in the paper to better position our work with respect to NOS. Thank you again for your time and feedback.

---

### Note · Authors · 2025-08-12

We thank the reviewers and AC for a constructive discussion. Per the committee’s intent, we use these final remarks to aid the decision by summarizing what was clarified.
During the discussion, we added context and results:
- **Additional results.** To show generality, we added molecule-generation results under instability constraints that prevent three-membered heterocycles. For language tasks, we added entropy metrics showing CDD does not ``game’' perplexity (some baselines do).
We justified that the counting-constraint implementation does not leak labels. We also clarified that rejection sampling is included as an implicit baseline and will be made explicit in the final version.
- **Novelty.** CDD performs a full primal–dual augmented-Lagrangian projection at each step, unlike guidance methods such as NOS. We emphasized the different mathematical framework and our feasibility guarantees.
- **Surrogate training.** We provided empirical evidence (perfect constraint satisfaction throughout, no gain from noisy-data training) and theoretical support (Theorem 4.1) for training neural surrogates, and linked this to training-free guidance work.
- **Notation.** We clarified definitions (e.g., $y^*$, Eq. 5). While the paper already defined these, we will further improve presentation.

**Planned revisions based on reviewer suggestions:**
1. **Method clarifications.** Expand Related Work to contrast CDD and NOS; highlight the primal–dual optimization. Add a clearer definition of $\beta$-prox-regularity; reposition notation (e.g., $y^*$ near Eq. 5). Move concrete score matching to preliminaries, and expand MDLM vs. UDLM.
2. **Structure.** Split $\S$4 into subsections, move Limitations into the main text, and improve signposting to appendix results.
3. **Experiments.** Include the new molecule-generation results reported with instability constraints and the entropy metric to show perplexity is not gamed. Expand the counting-experiment description,
4. **Generalization and scope.** Discuss generalization to unseen constraints; the only requirement is a differentiable constraint function (no task-specific projectors). Clarify on clean-data surrogate training and outline possible acceleration strategies. Note concurrent work scaling discrete diffusion LMs.

This summary is brief by design; the final version will incorporate the full discussion. Once again, thank you for the feedback and will address each concern in the revision.

Many thanks!
\
-The Authors

---

### Decision · Program_Chairs · 2025-09-17

**Decision:**

Accept (poster)

**Comment:**

This paper introduces a novel way to enforce constraints in discrete diffusion models using an augmented Lagrangian method. While the exact methodology is not technically novel, and it is questionable to what sequence lengths and vocabulary sizes it can be used for because of the Gumbel-softmax operation, the current results are compelling. Getting 100% constraint satisfaction while maintaining diversity is a good benchmark and should be applauded. The reviewers had several concerns regarding the clarity of presentation in sections 3-4 and some Appendix items that I encourage the authors to incorporate further. Furthermore, I encourage the authors to incorporate the feedback on related work and positioning of this paper, especially with respect to the growing body of work on finetuning discrete diffusion models. Having said that, I still think this is an interesting contribution that should be presented at NeurIPS.